# Learning Layer-wise Equivariances Automatically using Gradients

**Tycho F.A. van der Ouderaa**[1]       **Alexander Immer**[2,3]       **Mark van der Wilk**[1,4]

[1]Department of Computing, Imperial College London, United Kingdom
[2]Department of Computer Science, ETH Zurich, Switzerland
[3]Max Planck Institute for Intelligent Systems, Tübingen, Germany
[4]Department of Computer Science, University of Oxford, United Kingdom

## Abstract

Convolutions encode equivariance symmetries into neural networks leading to better generalisation performance. However, symmetries provide fixed hard constraints on the functions a network can represent, need to be specified in advance, and can not be adapted. Our goal is to allow flexible symmetry constraints that can automatically be learned from data using gradients. Learning symmetry and associated weight connectivity structures from scratch is difficult for two reasons. First, it requires efficient and flexible parameterisations of layer-wise equivariances. Secondly, symmetries act as constraints and are therefore not encouraged by training losses measuring data fit. To overcome these challenges, we improve parameterisations of soft equivariance and learn the amount of equivariance in layers by optimising the marginal likelihood, estimated using differentiable Laplace approximations. The objective balances data fit and model complexity enabling layer-wise symmetry discovery in deep networks. We demonstrate the ability to automatically learn layer-wise equivariances on image classification tasks, achieving equivalent or improved performance over baselines with hard-coded symmetry.

## 1 Introduction

Symmetry constraints, such as layer-wise equivariances of convolutional layers, allow neural networks to generalise efficiently [Cohen and Welling, 2016]. However, it is not always clear which symmetries to use and how much they should be enforced. In digit recognition, some rotational robustness is desirable, but strict invariance may make 6's and 9's indistinguishable. Similarly, translation equivariance prevents rate coding of relevant positional information [Sabour et al., 2017]. We propose a method to automatically learn the type and amount of layer-wise equivariances from training data.

To do so, we relax symmetries building upon recent literature on approximate equivariance, including residual pathways [Finzi et al., 2021a] and non-stationary filters [van der Ouderaa et al., 2022], effectively allowing controllable interpolation between strict equivariance or less constrained functions. To remain practical, we propose extensions to keep the parameter count similar to classic convolutions.

Allowing for differentiable selection between equivariances alone is not enough, as symmetries act as constraints and are therefore not encouraged by training losses that measure data fit, as noted in van der Ouderaa and van der Wilk [2021], Immer et al. [2021]. Search over many continuous hyperparameters with cross-validation is infeasible, relying on expensive retraining, hold-out validation data and does not leverage gradients. Instead, we follow Bayesian model selection through differentiable Laplace approximations to learn layer-wise equivariances from training data in a single training procedure.

We demonstrate automatically learning layer-wise symmetry structure on image classification tasks. To do so, we improve upon existing parameterisations of differentiable equivariance and derive corresponding Kronecker-factored Laplace approximations to the marginal likelihood. On image classification, we show that our method automatically learns convolutional structure in early layers and achieves similar or improved performance compared to architectures with hard-coded symmetry.

---

Code accompanying this paper is available at `https://github.com/tychovdo/ella`
37th Conference on Neural Information Processing Systems (NeurIPS 2023).

## 2 Related Work

**Enforcing strict equivariance into the architecture.** It can be shown that enforcing equivariance constraints in linear neural network layers is equivalent to performing a convolution [Kondor and Trivedi, 2018, Cohen et al., 2019]. Regular convolutional layers embed equivariance to translation, but this notion can be extended to other groups by convolving over groups [Cohen and Welling, 2016]. Group convolutions have been proposed for various domains and more complex continuous groups [Weiler et al., 2018, Weiler and Cesa, 2019, Worrall et al., 2017]. Equivariances enforce hard constraints on functions and can even form a linear subspace that can be explicitly computed [Van der Pol et al., 2020, Finzi et al., 2021b].

**Place coding, rate coding, disentanglement and part-whole hierarchies.** Equivariance closely relates to disentangled representations between place-coded and rate-coded features found in capsule networks [Sabour et al., 2017, Hinton et al., 2018, Kosiorek et al., 2019], as formalised in Cohen and Welling [2014], Cohen et al. [2018]. Positional information under strict translation equivariance of convolutions can only be place-coded, not rate-coded in features. Capsule networks aim to break this strict disentanglement to allow the embedding of part-whole hierarchies [Sabour et al., 2017]. We argue that our method solves the same problem by relaxing the disentanglement of strict equivariance and provide an automatic procedure to learn the extent to which information is exchanged and moved from place-coded to rate-coded features.

**Approximate equivariance.** Symmetries are often assumed known, must be fixed, and can not be adapted. This can be desirable if symmetries are known [Veeling et al., 2018], but becomes overly restrictive in other real-world tasks where symmetries or the appropriate amount of symmetry is not clear. Enforcing symmetry regardless can hamper performance [Liu et al., 2018b, Wang et al., 2022].

Approximate symmetry can prevent symmetry misspecification without losing the benefits that inductive biases of symmetries provide. Although strict group symmetries always apply to the entire group, by the group theoretical axiom of closure, approximate symmetry can be thought of as a form of robustness towards group actions that is only locally enforced (around training data). We use relaxations of symmetry that follow notions of approximate symmetry in literature [van der Wilk et al., 2018, Wang et al., 2020]. Although invariances can often easily be relaxed by simply averaging data augmentations [Benton et al., 2020, Schwöbel et al., 2021, van der Ouderaa and van der Wilk, 2021], parameterising approximate equivariance [Wang et al., 2022] can be more difficult. We build upon recent attempts, including partial equivariance [Romero and Lohit, 2021], residual pathways [Finzi et al., 2021a], and non-stationary filters [van der Ouderaa et al., 2022, van der Ouderaa and van der Wilk, 2023]. Existing relaxations of equivariance often introduce many additional parameters due to added linear layers [Finzi et al., 2021a] or use of hyper networks [Romero and Lohit, 2021, van der Ouderaa et al., 2022], making adoption in practice difficult. We strive to keep parameter counts close to classical convolutional layers, sometimes requiring further simplifying assumptions.

**Symmetry discovery and objectives** To learn symmetry, we relax layer-wise symmetries and thereby allow them to be differentiably learned with gradients, with strict symmetry as limiting case. Since adding symmetries does not directly improve training losses that rely on training fit, there is no direct encouragement of the objective function to use symmetries. To overcome this, some works have considered using validation data [Maile et al., 2022, Zhou et al., 2020]. Alternatively, Benton et al. [2020], Finzi et al. [2021a], Yang et al. [2023], van der Ouderaa et al. [2022] consider learning symmetry from training data only, but to do so require explicit regularisation that needs tuning. Some problems with this approach have been noted in Immer et al. [2022], which, like van der Wilk et al. [2018], proposes to use Bayesian model selection to learn invariances in a single training procedure, use differentiable Laplace approximations to remain tractable in deep neural networks. The method only considers invariances, which can be less efficient because it relies on learning parameters in the sampler instead of learning regularized symmetry in intermediary representations. We extend this line of work to enable automatic learning of layer-wise equivariances from training data.

**Neural architecture search** As layer-wise equivariances imply a convolutional architectural structure [Cohen et al., 2019], symmetry discovery can be viewed as a form of neural architecture search (NAS). Our approach uses relaxed symmetries to allow differentiable search over architectural structures, similar to DARTS [Liu et al., 2018a]. Instead of using validation data, however, our method uses Bayesian model selection to learn architecture from training data only.

We distinguish ourselves from large parts of AutoML literature in that we do not require expensive outer loops [Zoph et al., 2018], external agents [Cubuk et al., 2018], or even validation data [Lorraine et al., 2020, Yeh et al., 2022]. Our method can be interpreted as a one-shot architecture search using Laplace approximations, similar to [Zhou et al., 2019]. We demonstrate equivariance learning, but see no theoretical reason why the proposed objective should be limited to symmetry discovery, making it potentially broadly applicable in architecture search. For instance, in approaches that aim to learn network depth [Antorán et al., 2020], filter sizes [Romero and Zeghidour, 2023] and other layer types from training data. Although interesting future work, we focus on learning layer-wise equivariances.

**Bayesian Model Selection**    We propose to treat layer-wise equivariances as hyperparameters and learn them with Bayesian model selection – a well-understood statistical procedure. Traditionally, model selection has only been available for models with tractable marginal likelihoods, such as Gaussian Processes [Rasmussen, 2003], but approximations for deep learning have also been developed, such as Variational Inference schemes [Ober and Aitchison, 2021], deep Gaussian Processes [Dutordoir et al., 2020], and scalable Laplace approximations [Immer et al., 2021], which we use. Although application in deep learning involves additional approximations, the underlying mechanism is similar to classic Automatic Relevance Determination [MacKay et al., 1994] to optimise exponential basis function lengthscales. Even more so, as we cast layer-wise equivariance learning as a lengthscale selection problem in Section 3.2. Learning invariances with Bayesian model selection in GPs was demonstrated in van der Wilk et al. [2018], and extensions to neural networks exist [Schwöbel et al., 2021, van der Ouderaa and van der Wilk, 2021], but often use lower bounds that only work on shallow networks. For deep networks, Immer et al. [2022, 2023] and Mlodozeniec et al. [2023] recently demonstrated invariance learning with marginal likelihood estimates in large ResNets.

This work describes a method that enables layer-wise learning of equivariance symmetries from training data. We formulate different priors consisting of hyperparameters that place a different mass on functions that obey equivariance symmetry constraints and functions that do not. The amount of symmetry becomes a hyperparameter we empirically learn from data using Bayesian model selection.

In Section 3, we discuss how to efficiently parameterise relaxed equivariance in convolutional architectures. We propose additional factorisations and sparsifications in Section 3.1 to bring the parameter count of relaxed equivariance closer to that of classical convolutions. In Section 3.2, we describe how the parameterisations can be used to specify the amount of equivariance in the prior.

In Section 4, we deal with the objective function and discuss approximate marginal likelihood for layer-wise relaxed equivariance. We rely on recent advancements in scalable linearised Laplace approximations that use KFAC approximations of the Hessian, and in Sections 5.1 and 5.2 derive KFAC for proposed relaxed equivariance layers to use them in conjunction with this objective.

## 3    Differentiable parameterisation of layer-wise equivariance symmetry

To learn symmetries from training data, we use the strategy of relaxing equivariance constraints through parameterisations allowing interpolation between general linear mappings and linear mappings that are constrained to be strictly equivariant. In our choice of relaxation, we strive to keep parameter counts as close as possible to classical convolutional layers to remain useful in practice. We describe parameterisations of linear, equivariant and relaxed equivariant layers for discrete 2-dimensional translation symmetry $\mathbb{Z}^2$, as this is the most widely used symmetry in deep learning. All layers in this work can be naturally extended to other symmetry groups, as described in App. A.

**Fully-connected layer**    A fully-connected layer forms a linear map between an input feature map $\mathbf{x} : [0, C] \times \mathbb{Z}^2 \to \mathbb{R}$ (spatially supported at intervals $[0, X] \times [0, Y]$), with $C$ input channels outputting a $C'$ channel feature map $\mathbf{y} : [0, C'] \times \mathbb{Z}^2 \to \mathbb{R}$ with the same spatial support:

$$\mathbf{y}(c', x', y') = \sum_c \sum_{x,y} \mathbf{x}(c, x, y) \mathbf{w}(c', c, x', y', x, y) \tag{1}$$

where non-zero weights in $\mathbf{w} : \mathbb{Z}^6 \mapsto \mathbb{R}$ can be represented in $C'CX^2Y^2$ parameters. This may be more clear if, instead of using real-valued functions to describe features and weights, we equivalently flatten $\mathbf{x}, \mathbf{y}$ as vectors $\boldsymbol{y} = \boldsymbol{W}\boldsymbol{x}$ and write $\mathbf{w}$ as a matrix $\boldsymbol{W} \in \mathbb{R}^{C'XY \times CXY}$. Yet, we stick to the form of Eq. (1) as this simplifies future factorisations and generalisations. Furthermore, the notation emphasises that other layers, such as convolutions, form special cases of a fully-connected layer.

**Convolutional layer**  We might want to constrain a layer to be *strict equivariant*, so that translations in the input $(dx, dy) \in \mathbb{Z}^2$ yield equivalent translations in the output, mathematically:

$$\mathbf{y}(c', x' + dx, y' + dy) = \sum_c \sum_{x,y} \mathbf{x}(c, x + dx, y + dy)\mathbf{w}(c', c, x', y', x, y) \quad \forall dx, dy \in \mathbb{Z} \quad (2)$$

It can be shown that equivariance is equivalent (if and only if) to having a weight matrix that is *stationary* along the spatial dimension, meaning it solely depends on relative coordinates $\bar{x} = x' - x$ and $\bar{y} = y' - y$ and can thus be written in terms of stationary weights $\bar{\boldsymbol{\theta}} : (c, c', \bar{x}, \bar{y}) \mapsto \mathbb{R}$:

$$\mathbf{y}(c', x', y') = \sum_c \sum_{x,y} \mathbf{x}(c, x, y)\bar{\boldsymbol{\theta}}(c', c, x' - x, y' - y), \quad (3)$$

with $C'CXY$ parameters in $\bar{\boldsymbol{\theta}}$. Note that Eq. (3) is exactly a convolutional layer ('convolution is all you need', Theorem 3.1 of Cohen et al. [2019]). Constraining the stationary weights $\bar{\boldsymbol{\theta}}$ to be only locally supported on a rectangular domain $\bar{x}, \bar{y} \in [-\frac{S-1}{2}, \frac{S-1}{2}]$, and 0 otherwise, gives rise to small (e.g. $3 \times 3$) convolutional filters and further reduces the required parameters to $C'CS^2$.

### 3.1   Factoring and sparsifying relaxed equivariance

To learn layer-wise equivariances, we take the approach of relaxing symmetry constraints in neural network layers allowing explicit differentiable interpolation between equivariant and non-equivariant solutions. As a starting point, we consider the residual pathways of [Finzi et al., 2021a], reinterpreted as factorisation of the linear map as $\mathbf{w} = \boldsymbol{\theta} + \bar{\boldsymbol{\theta}}$ with fully-connected $\boldsymbol{\theta} : (c', c, x', y', x, y) \mapsto \mathbb{R}$ and stationary $\bar{\boldsymbol{\theta}}(c', c, \bar{x}, \bar{y}) \mapsto \mathbb{R}$ with $\bar{x} = x' - x$ and $\bar{y} = y' - y$:

$$\mathbf{y}(c', x', y') = \underbrace{\sum_c \sum_{x,y} \mathbf{x}(c, x, y)\boldsymbol{\theta}(c', c, x', y', x, y)}_{\text{fully-connected (FC)}} + \underbrace{\sum_c \sum_{x,y} \mathbf{x}(c, x, y)\bar{\boldsymbol{\theta}}(c', c, x' - x, y' - y)}_{\text{convolutional layer (CONV)}} \quad (4)$$

The layer consists of simply adding FC and CONV layer outputs. The CONV path spans the linear subspace of FC that obeys strict equivariance of Eq. (2). As such, the combined layer is equally expressive as a linear map (we can construct a bijection between $\mathbf{w}$ and $\boldsymbol{\theta} + \bar{\boldsymbol{\theta}}$, as is common for relaxed equivariance [van der Ouderaa et al., 2022]). In Finzi et al. [2021a], independent priors are placed on both sets of weights with relative variances controlling how much equivariance is encouraged. We also consider this prior, but assume it is unknown and use Bayesian model selection to automatically learn the variances that control symmetry constraints from data. Further, naive residual pathways of Eq. (4) require polynomially many parameters $C'CX^4 + C'CX^2$, or $C'CX^4 + C'CS^2$ for $S \times S$ filters, using square features $X = Y$ for simplicity. This makes using residual pathways in common architectures infeasible (see line 3 in Table 2). We overcome this by proposing factorisations and sparsifications bringing parameter counts of relaxed equivariance closer to convolutional layers.

**Factored layers**  Inspired by common factorisations in convolutional architectures, such as depth-wise [Chollet, 2017] and group separability [Knigge et al., 2022], we propose to factor fully-connected layers between input and output dimensions $\boldsymbol{\theta}(c', c, x', y', x, y) = \boldsymbol{\theta}_1(c', c, x', y')\boldsymbol{\theta}_2(c', c, x, y)$. This greatly reduces the parameters, while fully maintaining the benefits of fully-connected structure from full spatial $H \times W$ input and output $H' \times W'$ to both input and outputs channels $C \times C'$.

$$\sum_c \sum_{x,y} \mathbf{x}(c, x, y)\boldsymbol{\theta}(c', c, x', y', x, y) = \underbrace{\sum_c \boldsymbol{\theta}_1(c', c, x', y') \sum_{x,y} \mathbf{x}(c, x, y)\boldsymbol{\theta}_2(c', c, x, y)}_{\text{factored fully-connected (F-FC)}} \quad (5)$$

The factorisation of Eq. (5) has several benefits. First, it reduces the number of parameters required for the expensive the fully-connected path from $C'CX^2Y^2$ to $2C'CXY$. This is essential in practice to obtain parameter counts close to convolutional layers (133.6M→1.5M in Table 2). Secondly, we can write the layer as a composition of two linear operations summing over $\sum_{x,y}$ and $\sum_c$, which we use to derive Kronecker-factored approximations to the second-order curvature and enable the use of scalable Laplace approximations. Lastly, the parameters $\boldsymbol{\theta}_1$ and $\boldsymbol{\theta}_2$ have a single spatial domain that lends themselves to further optional sparsification with basis functions.

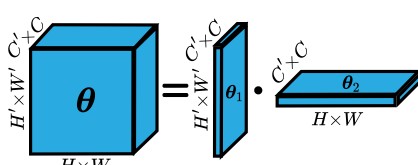

Figure 1: Eq. (5) visualised as tensors.

**Spatial sparsification** To further improve the parameter efficiency, we consider parameterising spatial dimensions in a lower dimensional basis $\bar{s}_{\boldsymbol{\theta}}^{\omega}(c', c, \bar{x}, \bar{y}) = \sum_{j=1}^{M} a_j \phi_j^{c',c}(\bar{x}, \bar{y})$. Our approach is partly inspired by B-spline Lie group filters of [Bekkers, 2019]. We follow van der Ouderaa and van der Wilk [2023] proposing

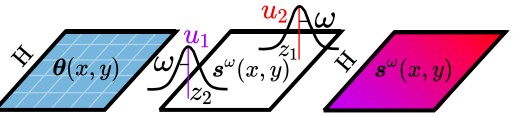

Figure 2: Spatial sparsification $s^{\omega}$ illustrated.

standard exponential basis functions $\phi_j^{c',c}(\bar{x}, \bar{y}) = u_j \exp(-\omega^2([\bar{x} \quad \bar{y}]^T - z_j)^2)$ for convolutional filters, with anchor point locations $z_j \in \mathbb{R}^2$ and anchor point values $u_j \in \mathbb{R}$. The work demonstrated that lengthscales $\omega$ can allow differentiable control over equivariances (used in Section 3.2) as well as practical sparsification with negligible performance loss, even if filters are already small (e.g. $3 \times 3$).

$$\sum_c \sum_{x,y} \mathbf{x}(c,x,y) \bar{\boldsymbol{\theta}}(c', c, x'-x, y'-y) = \underbrace{\sum_c \sum_{x,y} \mathbf{x}(c,x,y) \bar{s}^{\omega}(c', c, x'-x, y'-y)}_{\text{sparse convolution (S-CONV)}} \quad (6)$$

This reduces the number of parameters from $C'CS^2$ to $C'CP$ with $P$ anchor points, or $C'CP + 2P$ if anchor point locations $\{z_j\}_{j=1}^M$ (shared over channels) are also considered learnable parameters. As weights in our proposed factored F-FC have independently factored spatial input and output domains, we can perform a similar sparsification. We write $s_1^{\omega}$ and $s_2^{\omega}$ for spatially sparsified $\boldsymbol{\theta}_1$ and $\boldsymbol{\theta}_2$:

$$\sum_c \boldsymbol{\theta}_1(c', c, x', y') \sum_{x,y} \mathbf{x}(c,x,y) \boldsymbol{\theta}_2(c', c, x, y) = \underbrace{\sum_c s_1^{\omega}(c', c, x', y') \sum_{x,y} \mathbf{x}(c,x,y) s_2^{\omega}(c', c, x, y)}_{\text{sparse fully-connected (S-FC)}} \quad (7)$$

reducing the parameter cost from $2C'CXY$ to $C'CP$ with $P$ total number of anchor points. Compared to convolutional filters, which already have small spatial supports, sparsification of fully-connected layers will typically result in a much larger absolute reduction of parameters, for fixed $P$. As discussed in App. A, sparsified layers of Eqs. (6) and (7) can be extended to other groups, but require well-defined basis functions $\phi$ on groups, as explored in [Azangulov et al., 2022].

## 3.2 Specifying the amount of equivariance in the prior

To allow for learnable equivariance, we explicitly specify symmetry constraints in the prior and then empirically learn them through approximate Bayesian model selection. We choose a Gaussian form $\mathcal{N}(\mathbf{0}, \sigma_l^2 \mathbf{I})$ and treat variances as hyperparameters $\boldsymbol{\eta} = [\sigma_1, \sigma_2, \ldots]$. We carefully parameterise the network, such that the prior in each layer $l$ enforces an equivariance symmetry in the following limit:

$$\sigma_l^2 = 0 \implies \text{strict equivariance (Eq. (2))} \quad \text{and} \quad \sigma_l^2 > 0 \implies \text{relaxed equivariance} \quad (8)$$

We can think of different possible ways to parameterise neural network layers such that the conditions of Eq. (8) are met. For instance, Finzi et al. [2021a] propose placing priors directly on the weights of both residual pathways FC+CONV $\mathcal{N}(\boldsymbol{\theta}|\mathbf{0}, \sigma_l^2 \mathbf{I})$ and $\mathcal{N}(\bar{\boldsymbol{\theta}}|\mathbf{0}, \bar{\sigma}_l^2 \mathbf{I})$ with scalar variances $\sigma_l^2, \bar{\sigma}_l^2 \in \mathbb{R}$ and identity matrix $\mathbf{I} \in \mathbb{R}^{|\boldsymbol{\theta}| \times |\boldsymbol{\theta}|}$. Intuitively, all weights in non-equivariant paths become zero in the zero variance limit $\sigma_l = 0$, resulting in strict equivariance. We consider the same prior for the factored parameterisation F-FC+CONV. Likewise, for sparsified layers, we can place the prior directly on anchor points $\mathcal{N}(\boldsymbol{u}|\mathbf{0}, \sigma_l^2 \mathbf{I})$ and $\mathcal{N}(\bar{\boldsymbol{u}}|\mathbf{0}, \bar{\sigma}_l^2 \mathbf{I})$ which will control equivariance through a similar mechanism. For S-CONV, this prior corresponds to the correlated prior of Fortuin et al. [2021] if anchor points are placed exactly on the filter grid, whereas our anchor points are sparse $M < S^2$ and have learnable positions $z_j$. Alternatively, for sparsified S- layers, we can consider placing the prior directly on the lengthscales $\mathcal{N}(\omega|\mathbf{0}, \sigma_l^2 \mathbf{I})$ and $\mathcal{N}(\bar{\omega}_l|\mathbf{0}, \bar{\sigma}_l^2 \mathbf{I})$, as proposed in non-stationary filters [van der Ouderaa et al., 2022, van der Ouderaa and van der Wilk, 2023]. Intuitively, the limit of $\sigma_l = 0$ forces $\omega_l$ to be zero, causing non-stationary components to be constant and weights to act stationary, resulting in strict equivariance. Treating lengthscales $\omega$ as learnable hyperparameter has the additional advantage that it allows learning of convolutional filter frequencies from data through model selection of hyperparameter $\bar{\sigma}_l^2$.

Different constructions of the prior may induce different prior probabilities in function space. Nevertheless, it can be shown that Eq. (8) holds for all choices of prior described above. Thus, we have explicit control over symmetry with strict equivariance at the $\sigma_l = 0$ limiting case and relaxed layer-wise equivariance at higher variances $\sigma_l > 0$. Unlike prior works [Finzi et al., 2021a, van der Ouderaa et al., 2022] that require setting or tuning the prior variances $\sigma_l^2$ controlling equivariance constraints, we follow Bayesian model selection to learn them automatically from training data.

## 4 Bayesian model selection of symmetry constraints

Inspired by Bayesian methodology, we propose to empirically learn the amount of layer-wise equivariance in a neural network from training data through approximate Bayesian model selection. We define equivariances in our prior and treat the amount of symmetry as hyperparameters optimised using the marginal likelihood – a well-understood statistical procedure known as 'empirical Bayes'. Mathematically, we find a point estimate of the hyperparameters $\boldsymbol{\eta}_* = \arg\max_{\boldsymbol{\eta}} p(\mathcal{D}|\boldsymbol{\eta})$, whilst integrating out the $P$ model parameters $\boldsymbol{\theta} \in \mathbb{R}^P$:

$$p(\mathcal{D}|\boldsymbol{\eta}) = \int_{\boldsymbol{\theta}} p(\mathcal{D}|\boldsymbol{\theta}, \boldsymbol{\eta}) p(\boldsymbol{\theta}) \mathrm{d}\boldsymbol{\theta}$$

The *marginal likelihood* $p(\mathcal{D}|\boldsymbol{\eta})$ forms the normalising constant of the unnormalised posterior over parameters:

$$p(\boldsymbol{\theta}|\mathcal{D}, \boldsymbol{\eta}) = \frac{p(\mathcal{D}|\boldsymbol{\theta}, \boldsymbol{\eta}) p(\boldsymbol{\theta})}{p(\mathcal{D}|\boldsymbol{\eta})} = \frac{p(\mathcal{D}|\boldsymbol{\theta}, \boldsymbol{\eta}) p(\boldsymbol{\theta})}{\int_{\boldsymbol{\theta}} p(\mathcal{D}|\boldsymbol{\theta}, \boldsymbol{\eta}) p(\boldsymbol{\theta}) \mathrm{d}\boldsymbol{\theta}}$$

As the marginal likelihood is constant in $\boldsymbol{\theta}$, it can be ignored when training neural network parameters with a regular MAP point-estimate $\boldsymbol{\theta}_* = \arg\min_{\boldsymbol{\theta}} \mathcal{L}_{\boldsymbol{\theta}}$:

$$\mathcal{L}_{\boldsymbol{\theta}} = -\log p(\mathcal{D}|\boldsymbol{\theta}, \boldsymbol{\eta}) - \log p(\boldsymbol{\theta})$$

To optimise hyperparameters, the marginal likelihood $p(\mathcal{D}|\boldsymbol{\eta})$ can not be ignored and becomes important again as the objective function for hyperparameters. The marginal likelihood is the integral over the very high-dimensional space of possible parameter configurations, which is intractable for large neural networks and therefore requires additional approximations in practice.

To overcome the intractability of the marginal likelihood, the Laplace approximation [MacKay, 2003] can be used to estimate the normalising constant by first approximating the posterior by a Gaussian centered at the mode $\boldsymbol{\mu} = \boldsymbol{\theta}_*$ and the covariance set to the local curvature $\boldsymbol{\Sigma} = \mathbf{H}_{\boldsymbol{\theta}_*}^{-1} := -\nabla_{\boldsymbol{\theta}}^2 \mathcal{L}_{\boldsymbol{\theta}}|_{\boldsymbol{\theta}=\boldsymbol{\theta}_*}$ at that point:

$$p(\boldsymbol{\theta}|\mathcal{D}, \boldsymbol{\eta}) \approx q(\theta|\mathcal{D}, \boldsymbol{\eta}) = \mathcal{N}(\boldsymbol{\mu}, \boldsymbol{\Sigma})$$

Integrals of Gaussians are known in closed-form, and we can thus use the normalising constant of $q$ to estimate the log marginal likelihood objective used to optimise hyperparameters $\boldsymbol{\eta}_* = \arg\min_{\boldsymbol{\eta}} \mathcal{L}_{\boldsymbol{\eta}}$:

$$\mathcal{L}_{\boldsymbol{\eta}} = -\log\left(\sqrt{\frac{(2\pi)^P}{|\mathbf{H}_{\boldsymbol{\theta}_*}|}} p(\mathcal{D}, \boldsymbol{\theta}_*|\boldsymbol{\eta})\right) = \underbrace{-\log p(\mathcal{D}|\boldsymbol{\theta}_*, \boldsymbol{\eta})}_{\text{NLL / Data fit}} \underbrace{-\log p(\boldsymbol{\theta}_*) - \frac{P}{2}\log(2\pi) + \frac{1}{2}\log|\mathbf{H}_{\boldsymbol{\theta}_*}|}_{\text{Occam's factor}}$$
(9)

Although the Laplace approximation greatly simplifies the marginal likelihood, the Hessian determinant $|\mathbf{H}_{\boldsymbol{\theta}_*}|$ remains difficult to compute for large neural networks due to quadratic scaling in parameter count $\mathcal{O}(|\boldsymbol{\theta}|^2)$. In practice, we therefore use a local linearisation [Bottou et al., 2018] resulting in the generalized Gauss-Newton (GGN) approximation, which we further approximate using a block-diagonal Kronecker-factored (KFAC) structure Martens and Grosse [2015] (see Section 5).

The objective of Eq. (9) penalises model complexity enforcing a form of Occam's razor [Rasmussen and Ghahramani, 2000]. Complexity control is important when learning symmetry constraints, as previous works [van der Ouderaa and van der Wilk, 2021, Immer et al., 2021] have shown that just using the regular training loss $\mathcal{L}_{\boldsymbol{\theta}}$ w.r.t. data augmentation parameters prefers collapse into a solution without symmetry constraints resulting in fully-connected structure. The marginal likelihood overcomes this issue and can learn symmetries from data, as shown in van der Wilk et al. [2018]. The Laplace approximated marginal likelihood $\mathcal{L}_{\boldsymbol{\eta}}$ can be viewed as a form of curvature-aware minimisation and has recently been used to learn invariances in large ResNets [Immer et al., 2022].

## 5 Kronecker-factored curvature of relaxed equivariance layers

In Eq. (9), we propose the Laplace approximated log marginal likelihood $\mathcal{L}_{\boldsymbol{\eta}}$ as objective to learn layer-wise equivariances from data. As Hessians scale quadratically in the number of parameters

$\mathcal{O}(|\boldsymbol{\theta}|^2)$, the loss becomes intractable for large networks. We use KFAC [Martens and Grosse, 2015] to approximate the Hessian of the log likelihood $\boldsymbol{H}_{\boldsymbol{\theta}_*}$ and have the Gauss-Newton approximation $\boldsymbol{H}$:

$$\boldsymbol{H}_{\boldsymbol{\theta}_*} \approx \boldsymbol{H} = \sum_n \boldsymbol{H}_n = \sum_n \boldsymbol{J}(\boldsymbol{x}_n)^\top \boldsymbol{\Lambda}(\boldsymbol{x}_n) \boldsymbol{J}(\boldsymbol{x}_n),$$

where the Jacobians of network outputs $\boldsymbol{f}$ with respect to parameters $\boldsymbol{\theta}$ for data points $\boldsymbol{x}_n$ are $[\boldsymbol{J}(\boldsymbol{x}_n)]_{k,p} = \frac{\partial f_k}{\partial \theta_p}(\boldsymbol{x}_n)$, and $[\boldsymbol{\Lambda}(\boldsymbol{x}_n)]_{k,g} = \frac{\partial \log p(y_n|\boldsymbol{x}_n;\boldsymbol{f}(\boldsymbol{x}_n;\boldsymbol{\theta}))}{\partial f_k \partial f_g}$ is the log likelihood Hessian, all evaluated at $\boldsymbol{\theta}_*$. KFAC is a block-diagonal approximation of the GGN, with individual blocks $\boldsymbol{H}_l = \sum_n \boldsymbol{H}_{n,l} = \sum_n \boldsymbol{J}_l(\boldsymbol{x}_n)^\top \boldsymbol{\Lambda}(\boldsymbol{x}_n) \boldsymbol{J}_l(\boldsymbol{x}_n)$ for each layer $l$ with parameters $\boldsymbol{\theta}_l$ and Jacobian $\boldsymbol{J}_l(\boldsymbol{x})$.

For a fully-connected layer, $\boldsymbol{\theta}\boldsymbol{x}$ with $\boldsymbol{\theta} \in \mathbb{R}^{G_l \times D_l}$ and $\boldsymbol{x} \in \mathbb{R}^{D_l}$, we can write Jacobians of a single data point w.r.t. parameters as $\boldsymbol{J}_l(\boldsymbol{x}_n)^\top = \boldsymbol{a}_{l,n} \otimes \boldsymbol{g}_{l,n}$, where $\boldsymbol{a}_{l,n} \in \mathbb{R}^{D_l \times 1}$ is the input to the layer and $\boldsymbol{g}_{l,n} \in \mathbb{R}^{G_l \times K}$ is the transposed Jacobian of $\boldsymbol{f}$ w.r.t. the $l$th layer output. We thus have:

$$\boldsymbol{H}_l = \sum_n \boldsymbol{J}_l(\boldsymbol{x}_n)^\top \boldsymbol{\Lambda}(\boldsymbol{x}_n) \boldsymbol{J}_l(\boldsymbol{x}_n) = \sum_n [\boldsymbol{a}_{l,n} \otimes \boldsymbol{g}_{l,n}] \boldsymbol{\Lambda}(\boldsymbol{x}_n) [\boldsymbol{a}_{l,n} \otimes \boldsymbol{g}_{l,n}]^\top$$
$$= \sum_n [\boldsymbol{a}_{l,n} \boldsymbol{a}_{l,n}^\top] \otimes [\boldsymbol{g}_{l,n} \boldsymbol{\Lambda}(\boldsymbol{x}_n) \boldsymbol{g}_{l,n}^\top] \approx \frac{1}{N} \left[ \sum_n \boldsymbol{a}_{l,n} \boldsymbol{a}_{l,n}^\top \right] \otimes \left[ \sum_n \boldsymbol{g}_{l,n} \boldsymbol{\Lambda}(\boldsymbol{x}_n) \boldsymbol{g}_{l,n}^\top \right].$$

KFAC only requires the computation of the two last Kronecker factors, which are relatively small compared to the full GGN block. KFAC has been extended to convolutional CONV layers in Grosse and Martens [2016]. We derive KFAC also for factored layers F-FC of Eq. (5) and sparsified S-FC and S-CONV layers of Eqs. (6) and (7), allowing use in scalable Laplace approximations to learn layer-wise equivariances from data. The essential step to extend KFAC is to write the Jacobian of the $l$th layer, $\boldsymbol{J}_l(\boldsymbol{x})$, as Kronecker product allowing data points to be factored and approximated with the last KFAC step. Our derivations of KFAC for new layers are in detail provided in Apps. C.1 and C.2.

## 5.1 KFAC for factored layers

To obtain the KFAC of F-FC in Eq. (5), we write the application of both weight parameters $\boldsymbol{\theta}_1$ and $\boldsymbol{\theta}_2$ as a matrix mutiplication akin to a linear layer, which allows us to apply a similar approximation as in standard KFAC. In particular, we first split up the operations in Eq. (5) into first applying $\boldsymbol{\theta}_2$ and then $\boldsymbol{\theta}_1$. Each of these operations can be written as a matrix product $\boldsymbol{\theta}\boldsymbol{x}$ with corresponding appropriate sizes of $\boldsymbol{\theta}$ and $\boldsymbol{x}$. In comparison to the standard KFAC case, we have a Jacobian that can be written as

$$\boldsymbol{J}_{\boldsymbol{\theta}}^\top = \sum_c \boldsymbol{x}_c \otimes [\nabla_{\mathbf{y}_c} \boldsymbol{f}]^\top = \sum_c \boldsymbol{a}_c \otimes \boldsymbol{g}_c,$$

which means an additional sum needs to be handled. In particular, plugging the Jacobian structure into the standard KFAC approximation, we have

$$\boldsymbol{H}_{\boldsymbol{\theta}} \approx \sum_n \boldsymbol{J}_{\boldsymbol{\theta}}(\boldsymbol{x}_n)^\top \boldsymbol{\Lambda}(\boldsymbol{x}_n) \boldsymbol{J}_{\boldsymbol{\theta}}(\boldsymbol{x}_n) = \sum_n [\sum_c \boldsymbol{a}_{n,c} \otimes \boldsymbol{g}_{n,c}] \boldsymbol{\Lambda}(\boldsymbol{x}_n) [\sum_c \boldsymbol{a}_{n,c} \otimes \boldsymbol{g}_{n,c}]^\top$$
$$= \sum_{n,c,c'} [\boldsymbol{a}_{n,c} \boldsymbol{a}_{n,c}^\top] \otimes [\boldsymbol{g}_{n,c'} \boldsymbol{\Lambda}(\boldsymbol{x}_n) \boldsymbol{g}_{n,c'}^\top] \approx \frac{1}{NC} [\sum_{n,c} \boldsymbol{a}_{n,c} \boldsymbol{a}_{n,c}^\top] \otimes [\sum_{n,c} \boldsymbol{g}_{n,c} \boldsymbol{\Lambda}(\boldsymbol{x}_n) \boldsymbol{g}_{n,c}^\top],$$

where the first approximation is from the Hessian to the GGN and the second approximation is due to KFACs exchange of sums and products in favour of numerical efficiency. The efficiency comes due to the fact that we have two Kronecker factors instead of a single dense matrix. This allows to compute and optimize the marginal likelihood approximation efficiently and therefore tune $\sigma_l$ values. In App. C.1, we derive the KFAC for $\boldsymbol{\theta}_1$ and $\boldsymbol{\theta}_2$ in detail.

## 5.2 KFAC for sparsified layers

To obtain KFAC for sparsified S-FC and S-CONV layers, we extend derivations of KFAC for respective factored F-FC layers of Section 5.1 and CONV layers in [Grosse and Martens, 2016]. To compute Jacobians with respect to our new anchor points $\boldsymbol{a}$ parameters, instead of the induced weights $\boldsymbol{s}^\omega$, we apply the chain rule:

$$\frac{\partial \boldsymbol{f}}{\partial a_j} = \frac{\partial \boldsymbol{f}}{\partial \boldsymbol{s}^\omega} \frac{\partial \boldsymbol{s}^\omega}{\partial a_j}, \text{ using the partial derivative} \qquad \frac{\partial \boldsymbol{s}^\omega(c', c, x, y)}{\partial a_j} = \phi_j^{c',c}(x,y)$$

Consequently, we can compute KFAC for sparsified layers by projecting the appropriate dimensions of Kronecker factors of existing KFAC derivations with basis functions $\phi$ stored in the forward pass. In App. C.2, we derive KFAC for sparsified S-FC and S-CONV layers in detail.

# 6 Experiments

## 6.1 Toy problem: adapting symmetry to task

The symmetries in a neural network should be task dependent. Imagine a task where the full symmetry would be too restrictive. *Can our method automatically adapt to what is required by the task?* Although this ability arguably becomes most important on more complex datasets where symmetries are not strict or completely unknown, the absence of ground-truth symmetry in such scenarios makes evaluation more difficult. We, therefore, turn to a simple toy problem of which underlying symmetries are known and consider more interesting larger-scale datasets in the following sections. We slightly altered the MNIST dataset [LeCun et al., 1989] so that digits are randomly placed in one of four image quadrants (see example left of Table 1) and consider the task of classifying the correct digit out of 10 classes, and the task of classifying both the digit and the quadrant it is placed in, totalling 40 classes. The first task is translation invariant, as moving a digit around does not change its class. The second task, on the other hand, can not be solved under strict translation invariance due to the dependence on location. As expected, we find in Table 1 that the CONV model outperforms the FC model on the first strictly symmetric task. Conversely, the CONV model fails to solve the second task due to the symmetry misspecification and is consequently outperformed by the FC model. The proposed method with adjustable symmetry achieves high test accuracy on both tasks.

| Example: | | | MAP | | Diff. Laplace | | |
|---|---|---|---|---|---|---|---|
| | | | FC | CONV | FC | CONV | Learned (**ours**) |
| | Symmetry | Prediction task | 112.7 M | 0.4 M | 112.7 M | 0.4 M | 1.8 M |
| | Strict symmetry | *(digit)* | 95.73 | 99.38 | 97.04 | 99.23 | 98.86 |
| *(3, upper left)* | Partial symmetry | *(digit, quadrant)* | 95.10 | 24.68 | 95.46 | 24.50 | 99.00 |

Table 1: Preventing symmetry misspecification. Test accuracy for non-symmetric FC, strictly symmetric CONV, and learnable symmetry F-FC+CONV models on a translation invariant task and a task that can not be solved under strict translation symmetry. Unlike the FC and CONV baselines, the proposed model with learnable symmetry constraints achieves high test performance on both tasks.

## 6.2 Learning to use layer-wise equivariant convolutions on CIFAR-10

Convolutional layers provide useful inductive bias for image classification tasks, such as CIFAR-10 [Krizhevsky et al., 2009]. We investigate whether our method can select this favourable convolutional structure when presented with this task. We follow Finzi et al. [2021a] and use the architecture of Neyshabur [2020] (see App. F). In Table 2, we compare test performance when using fully-connected FC, convolutional CONV, residual pathways FC+CONV, and the proposed more parameter-efficient factorisations F-FC and F-FC+CONV and sparsified S- layers. The prior is placed on the weights $\theta$, anchor points $u$, or lengthscales $\omega$, as discussed in Section 3.2. As expected, CONV layers perform best under regular MAP training, followed by less constrained FC and FC+CONV layers. If we, instead, use the Laplace approximated marginal likelihood estimates, we find that learnable symmetry obtains equivalent (within <1%) performance as the convolutional CONV models.

| Layer | Learnable equivariance | Prior $\mathcal{N}(\cdot\|\eta)$ | # Params | MAP | | Diff. Laplace (**Ours**) | | |
|---|---|---|---|---|---|---|---|---|
| | | | | Test NLL ($\downarrow$) | Test Acc. ($\uparrow$) | Test NLL ($\downarrow$) | Test Acc. ($\uparrow$) | $\mathcal{L}_\eta(\downarrow)$ |
| FC | | $\theta$ | 133.6 M | 2.423 | 64.42 | 1.319 | 52.36 | 1.896 |
| CONV | | $\theta$ | 0.3 M | **1.184** | **82.81** | **0.464** | **84.07** | **1.022** |
| FC+CONV | ✓ | $\theta$ | 133.9 M | 1.713 | 76.93 | **0.489** | 83.32 | **1.019** |
| F-FC | | $\theta$ | 1.5 M | 4.291 | 50.47 | 1.307 | 53.38 | 2.119 |
| F-FC+CONV (**Ours**) | ✓ | $\theta$ | 1.8 M | 1.343 | 81.61 | **0.468** | 83.92 | **1.277** |
| S-FC | | $u$ | 0.8 M | 5.614 | 50.21 | 1.496 | 46.18 | 1.926 |
| S-CONV | | $u$ | 0.2 M | **0.604** | **82.53** | **0.562** | 81.29 | **1.037** |
| S-FC+S-CONV (**Ours**) | ✓ | $u$ | 1.0 M | 5.485 | 58.80 | **0.538** | 81.55 | **0.978** |
| S-FC | | $\omega$ | 0.8 M | 7.783 | 49.69 | 1.319 | 52.22 | 1.813 |
| S-CONV | | $\omega$ | 0.2 M | **0.913** | **81.84** | **0.549** | 81.20 | **1.055** |
| S-FC+S-CONV (**Ours**) | ✓ | $\omega$ | 1.0 M | 2.040 | 77.97 | 0.564 | **80.63** | 1.067 |

Table 2: Equivariant convolutional layers CONV obtain the highest performance on image classification task when trained with regular MAP. When maximising Laplace approximated marginal likelihood, models with learnable symmetry learn from training data to 'become convolutional' (see Section 6.2 for a discussion) and obtain equivalent (within <1%) performance as CONV models.

**Layers learn to become convolutional, except at the end of the network.**
Upon further inspection of learned symmetries, we observe very high prior precisions $\sigma_l^{-2} > 10^6$ for non-equivariant paths in most layers, thus negligibly contributing to final outputs (see App. D.1). We, therefore, conclude that layers have *learned to become convolutional* from training data, which also explains why training learned equivariance with Laplace performs on par with CONV models in Table 2. We reach the same conclusions upon inspection of effective dimensions [MacKay, 1992] in App. D.2 and visualise learned symmetries based on the highest relative effective number of parameters in Fig. 3. If layers did not become convolutional, this always occurred at the end of the network. This is particularly interesting as it coincides with common practice in literature. Spatial dimensions in the used architecture are halved through pooling operations, reaching 1 with sufficient or global pooling, after which features become strictly invariant and there is no difference anymore between 1x1 CONV and FC layers. Convolutional architectures in literature often flatten features and apply FC layers before this point [LeCun et al., 1998], breaking strict symmetry and allowing some dependence in the output on absolute position. Our findings indicate that the method can discover such favourable

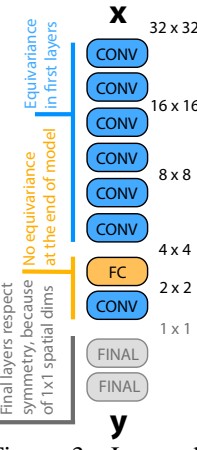

Figure 3: Learned CONV+FC model.

network designs automatically from data, outperforming regular MAP training. Furthermore, it shows that not always the same layer is selected, but different layer types are learned on a per-layer basis.

## 6.3 Selecting symmetry from multiple groups

App. A generalises the proposed method to other symmetry groups, and App. B describes layer-wise equivariance selection from a set of multiple symmetry groups or layer types. In this experiment, we consider adding discrete 90-degree rotations to the architecture F-FC+CONV+GCONV. In Table 3, we compare the test performance with MAP and approximated marginal likelihood training on versions of MNIST and CIFAR-10 datasets (details in App. F.5). We compensate for the added parameters resulting from the rotationally equivariant GCONV path by reducing channel sizes of individual paths by a factor of 5 ($\alpha=10$ to $\alpha=2$, see App. F). Still, we are able to obtain 80% test accuracy on CIFAR-10 when trained with approximated marginal likelihood. Upon analysis of learned prior variances (App. D.1) and effective number of parameters (App. D.2), we observe a positive correlation between learned symmetries and augmentations applied to the datasets trained on.

| | | MAP | | Learned with Differentiable Laplace (**ours**) | | | Rel. Effective Num. of Param. | | | | | |
| Dataset | # Params | Test NLL ($\downarrow$) | Test accuracy ($\uparrow$) | Test NLL ($\downarrow$) | Test accuracy ($\uparrow$) | Approx. MargLik ($\downarrow$) | FC (%) | | CONV (%) | | GCONV (%) | |
|---|---|---|---|---|---|---|---|---|---|---|---|---|
| MNIST | 1.2 M | 0.172 | 97.59 | 0.023 | **99.21** | 0.328 | 10 | (0-46) | 15 | (0-98) | 75 | (1-100) |
| Translated MNIST | 1.2 M | 0.812 | 90.78 | 0.053 | **98.27** | 0.216 | 0 | (0-0) | 23 | (0-99) | 77 | (1-100) |
| Rotated MNIST | 1.2 M | 0.819 | 91.02 | 0.136 | **95.55** | 0.896 | 8 | (0-20) | 8 | (0-47) | 83 | (47-100) |
| CIFAR-10 | 1.2 M | 3.540 | 68.33 | 0.552 | **80.94** | 0.926 | 0 | (0-1) | 44 | (0-99) | 56 | (0-100) |
| Rotated CIFAR-10 | 1.2 M | 5.953 | 48.30 | 1.236 | **55.68** | 1.630 | 4 | (0-22) | 14 | (0-41) | 82 | (58-99) |

Table 3: Selecting from multiple symmetry groups. Negative log likelihood (NNL) and Laplace learned symmetries measured by *mean (min-max)* relative effective number of parameters over layers.

## 7 Discussion and Conclusion

This work proposes a method to automatically learn layer-wise equivariances in deep learning using gradients from training data. This is a challenging task as it requires both flexible parameterisations of symmetry structure and an objective that can learn symmetry constraints. We improve upon existing parameterisations of relaxed equivariance to remain practical in the number of parameters. We learn equivariances through Bayesian model selection, by specifying symmetries in the prior and learning them by optimising marginal likelihood estimates. We derive Kronecker-factored approximations of proposed layers to enable scalable Laplace approximations in deep neural networks.

The approach generalises symmetry groups and can be used to automatically determine the most relevant symmetry group from a set of multiple groups. We rely on relaxing symmetries and learning the amount of relaxation, where strict equivariance forms a limiting case. In doing so, the method does require a list of symmetry groups and associated group representations that are considered, and learns to ignore, partially use, or strictly obey symmetry constraints. Yet, we argue this is a huge step forward compared to always using a single, strict symmetry. We hope that allowing artificial neural networks to automatically adapt structure from training data helps to leverage existing geometry in data better and reduce the reliance on trial-and-error in architecture design.

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
