# A  Generalisation to other groups

The main text describes layers with strict or relaxed equivariance to the translation group because this is the most commonly used symmetry in deep learning architectures used in classical convolutional layers. In this section, we show that all layers can be generalised to other symmetry groups $G$, where 2-dimensional discrete translations correspond to the special case of $G = \mathbb{Z}^2$. In this case any two group elements $(x, y) = g \in G = \mathbb{Z}^2$ and inverses become subtraction $g^{-1}g' = (x' - x, y' - y)$.

On groups, we consider input and output feature maps on a group $y : [0, B] \times [0, C] \times G$ and $x : [0, B] \times [0, C] \times G$, which can be achieved through group lifting [Kondor and Trivedi, 2018]. We generalise the linear layer of Eq. (1) on a group $G$ as:

$$\mathbf{y}(c', g') = \sum_c \sum_{g \in G} \mathbf{w}(c', c, g', g) \mathbf{x}(c, g)$$

Generalising regular convolutional layer to a group $G$ becomes the group equivariant convolution proposed in Cohen and Welling [2016]:

$$\mathbf{y}(c', g') = \sum_c \sum_g \mathbf{x}(c, g) \bar{\boldsymbol{\theta}}(c', c, g^{-1}g')$$

The residual pathway of Eq. (10) was originally in Finzi et al. [2021a] for general groups $G$. It can be seen as a factorisation $\mathbf{w} = \boldsymbol{\theta} + \bar{\boldsymbol{\theta}}$ where $\boldsymbol{\theta} : (c', c, g', g) \mapsto \mathbb{R}$ and $\bar{\boldsymbol{\theta}} : (c', c, g) \mapsto \mathbb{R}$:

$$\mathbf{y}(c', g') = \sum_c \sum_g \mathbf{x}(c, g) \boldsymbol{\theta}(c', c, g', g) + \sum_c \sum_g \mathbf{x}(c, g) \bar{\boldsymbol{\theta}}(c', c, g^{-1}g') \tag{10}$$

The factorisation of Eq. (5) proposed to further reduce the number of parameters generalises as:

$$\sum_c \sum_g \mathbf{x}(c, g) \boldsymbol{\theta}(c', c, g', g) = \sum_c \boldsymbol{\theta}_1(c', c, g') \sum_g \mathbf{x}(c, g) \boldsymbol{\theta}_2(c', c, g)$$

where $\boldsymbol{\theta}_1 : (c', c, g') \mapsto \mathbb{R}$ and $\boldsymbol{\theta}_2 : (c', c, g) \mapsto \mathbb{R}$. The sparse fully-connected layer of Eq. (7) generalises to groups $G$ as:

$$\sum_c \sum_g \mathbf{x}(c, g) \bar{\boldsymbol{\theta}}(c', c, g^{-1}g') = \sum_c \sum_g \mathbf{x}(c, g) \bar{\boldsymbol{s}}^\omega(c', c, g^{-1}g')$$

with stationary filters $\bar{\boldsymbol{\theta}} : (c', c, u) \mapsto \mathbb{R}$ taking group elements $u = g^{-1}g' \in G$ as argument. The sparse convolutional layer of Eq. (6) becomes:

$$\sum_c \boldsymbol{\theta}_1(c', c, g') \sum_g \mathbf{x}(c, g) \boldsymbol{\theta}_2(c', c, g) = \sum_c \boldsymbol{s}_1^\omega(c', c, g') \sum_g \mathbf{x}(c, g) \boldsymbol{s}_2^\omega(c', c, g)$$

where $\boldsymbol{s}_1^\omega : ...$ and $\boldsymbol{s}_2^\omega : ...$. In this case, the bases become functions on the group $\phi : G \to \mathbb{R}$, as have been explored in Azangulov et al. [2022].

# B  Selecting symmetry from multiple groups

Consider a set of $M$ groups $G_1, G_2, \ldots, G_M$, then we can factor multiple groups as follows:

$$\mathbf{y}(c', g') = \sum_{i=1}^M \sum_c \sum_{g \in G_i} \mathbf{x}(c, g) \boldsymbol{\theta}(c', c, g', g) + \sum_{i=1}^M \sum_c \sum_{g \in G_i} \mathbf{x}(c, g) \bar{\boldsymbol{\theta}}(c', c, g^{-1}g') \tag{11}$$

strictly generalising residual pathways of Eq. (10) to multiple groups $\{G_i\}_{i=1}^M$. The other layers can equivalently be extended to multiple groups.

Similar to the original residual pathway paper [Finzi et al., 2021a], we could also write linear mappings as matrix $\boldsymbol{W}$ and consider a set of groups $G_1, G_2, \ldots, G_M$ with representations and find a corresponding set of equivariant bases $\boldsymbol{B}_1, \boldsymbol{B}_2, \ldots \boldsymbol{B}_M$ [Finzi et al., 2021b], in which case we could write the layer of Eq. (11) equivalently as a sum of these linear maps:

$$\text{vec}(\boldsymbol{W}) = \sum_{i=1}^G \mathbf{B}_i \boldsymbol{u}_i + \boldsymbol{v}$$

The latter notation makes it possibly more clear that the layer forms a sum of linear layers that span subspaces associated with different equivariance constraints.

## C Extensions of Kronecker-factored approximate curvature (KFAC)

We briefly review KFAC for fully-connected neural networks and then extend it to the layers proposed in this work, i.e., factored layers and sparsified layers. Further, we give an extension for group convolutional layers. The goal of KFAC is to approximate the Hessian of the log likelihood, $H_{\theta_*}$, which is the overall loss Hessian from which the simple Hessian of the prior is subtracted. We have for the Gauss-Newton approximation $H$:

$$H_{\theta_*} \approx H = \sum_n H_n = \sum_n J(x_n)^\top \Lambda(x_n) J(x_n),$$

where $J(x_n)$ are the Jacobians of neural network output $f$ with respect to parameters $\theta$ for a given data point $x_n$, i.e., $[J(x_n)]_{k,p} = \frac{\partial f_k}{\partial \theta_p}(x_n)$, and $[\Lambda(x_n)]_{k,g} = \frac{\partial \log p(y_n|x_n; f(x_n;\theta))}{\partial f_k \partial f_g}$, which is the Hessian of the log likelihood with respect to the functional output of the neural network and is for common likelihoods independent of the label [Martens, 2020].

KFAC approximates the GGN in two efficient ways: first, the approximation is conducted in blocks, each corresponding to a single layer $l$, and, second, the GGN of each block is approximated as a Kronecker product. We denote a block corresponding to the $l$th layer as $H_l$, which is the GGN of the parameter $\theta_l$, and has the Jacobian $J_l(x)$. Then, we have

$$H_l = \sum_n H_{n,l} = \sum_n J_l(x_n)^\top \Lambda(x_n) J_l(x_n),$$

For a fully-connected layer, $\theta x$ with $\theta \in \mathbb{R}^{G_l \times D_l}$ and $x \in \mathbb{R}^{D_l}$, we can write the Jacobian of a single data point w.r.t. parameters as $J_l(x_n)^\top = a_{l,n} \otimes g_{l,n}$, where $a_{l,n} \in \mathbb{R}^{D_l \times 1}$ is the input to the layer and $g_{l,n} \in \mathbb{R}^{G_l \times K}$ is the transposed Jacobian of $f$ w.r.t. the output of the $l$th layer. KFAC can then be derived using the following equalities and lastly approximation:

$$H_l = \sum_n J_l(x_n)^\top \Lambda(x_n) J_l(x_n) = \sum_n [a_{l,n} \otimes g_{l,n}] \Lambda(x_n) [a_{l,n} \otimes g_{l,n}]^\top$$

$$= \sum_n [a_{l,n} a_{l,n}^\top] \otimes [g_{l,n} \Lambda(x_n) g_{l,n}^\top] \approx \frac{1}{N} \left[ \sum_n a_{l,n} a_{l,n}^\top \right] \otimes \left[ \sum_n g_{l,n} \Lambda(x_n) g_{l,n}^\top \right].$$

The resulting KFAC just requires to compute the two last Kronecker factors, which are relatively small compared to the full GGN block. To extend KFAC to the proposed layers, the essential step is to write the Jacobian of the $l$th layer, $J_l(x)$, as a Kronecker product so it can be factored for each data point and then approximated with the last KFAC step.

### C.1 KFAC for factored layers

Factored layers have two parameters $\theta_1, \theta_1$, each of which will result in a KFAC approximation of its respective curvature. First, we split up the definition from Eq. (5):

$$y(c', x', y') = \sum_c \sum_{x,y} x(c, x, y) \theta(c', c, x', y', x, y)$$

$$= \sum_c \theta_1(c', c, x', y') \sum_{x,y} x(c, x, y) \theta_2(c', c, x, y)$$

$$= \sum_c \theta_1(c', c, x', y') x_1(c', c),$$

which defines $x_1(c', c)$ as intermediate value, i.e., it is input to $\theta_1$ and output of operation $\theta_1$. Both individual operations can be equivalently written similar to linear layers by modifying the inputs.

For $\theta_2$, we write $\theta_2 \in \mathbb{R}^{c' \times cxy}$ and $x \in \mathbb{R}^{cxy \times c}$ which is given by repeating $x$ across the $c$ dimensions along the diagonal. We then have $x_1^{(o)} = \theta_2 x \in \mathbb{R}^{c' \times c}$. For $\theta_1$, we have similarly $\theta_1 \in \mathbb{R}^{x'y' \times cc'}$ and $x_1^{(i)} \in \mathbb{R}^{cc' \times c'}$, which is a block-diagonal constructed from the previous output $x_1^{(o)}$, i.e., $\theta_1$ is applied individually per $c'$. We have $y \in \mathbb{R}^{x'y' \times c'} = \theta_1 x_1^{(i)}$. To obtain KFAC for the two parameters, we need to write out the Jacobian as a Kronecker product of input data and output gradient. The

input data are given by the expanded $\boldsymbol{x}$ and transposed output Jacobians are $d\mathbf{y} \in \mathbb{R}^{x'y'c' \times K}$ and $d\boldsymbol{x} \in \mathbb{R}^{c'c \times K}$. This can be done as follows:

$$\boldsymbol{J}_{\boldsymbol{\theta}_2} = \sum_c d\boldsymbol{x}_c^\top \otimes \boldsymbol{x}_c, \qquad \text{and} \qquad \boldsymbol{J}_{\boldsymbol{\theta}_1} = \sum_{c'} d\mathbf{y}_{c'}^\top \otimes (\boldsymbol{x}_1^{(i)})_{c'}$$

where $c$ and $c'$ denote the index of the repeated dimension, respectively. Prototypically for $\boldsymbol{\theta}_2$, defining $\boldsymbol{a}_{l,n,c}$ as $\boldsymbol{x}_c$ and $\boldsymbol{g}_{l,n,c}$ as $d\boldsymbol{x}_c^\top$, we have the following KFAC approximation:

$$\boldsymbol{H}_l = \sum_n \left[ \sum_c \boldsymbol{a}_{l,n,c} \boldsymbol{g}_{l,n,c} \right] \otimes \left[ \sum_c \boldsymbol{a}_{l,n,c} \boldsymbol{\Lambda}(\boldsymbol{x}_n) \boldsymbol{g}_{l,n,c} \right]$$

$$\approx \frac{1}{NC} \sum_n \sum_c \left[ \boldsymbol{a}_{l,n,c} \boldsymbol{a}_{l,n,c}^\top \right] \otimes \left[ \boldsymbol{g}_{l,n,c} \boldsymbol{\Lambda}(\boldsymbol{x}_n) \boldsymbol{g}_{l,n,c}^\top \right],$$

which equivalently holds for $\boldsymbol{\theta}_1$ by replacing $c$ with $c'$, i.e., the number of input channels with output channels.

## C.2 KFAC for sparsified layers

For sparsified factored layers S-FC, we extend the K-FAC approximation for factored layers. We follow the same derivation as in App. C.1 to obtain Jacobians $\boldsymbol{J}_{\boldsymbol{s}_1^\omega}$ and $\boldsymbol{J}_{\boldsymbol{s}_2^\omega}$ with respect to weights induced by the basis functions $\boldsymbol{s}_1^\omega$, $\boldsymbol{s}_2^\omega$:

$$\boldsymbol{J}_{\boldsymbol{s}_1^\omega} = \sum_c d\boldsymbol{x}_c^\top \otimes \boldsymbol{x}_c, \qquad \text{and} \qquad \boldsymbol{J}_{\boldsymbol{s}_2^\omega} = \sum_{c'} d\mathbf{y}_{c'}^\top \otimes (\boldsymbol{x}_1^{(i)})_{c'}$$

In sparsified layers, $\boldsymbol{s}_1^\omega$ and $\boldsymbol{s}_2^\omega$ are functions and not part of the model parameters anymore. Instead, we are looking for the Jacobians $\boldsymbol{J}_{\boldsymbol{u}_1}$ and $\boldsymbol{J}_{\boldsymbol{u}_2}$ with respect to the sparser set of anchor point weights $\boldsymbol{u}_1$ and $\boldsymbol{u}_2$, which we can find by applying the chain rule:

$$\frac{\partial \boldsymbol{f}}{\partial \boldsymbol{u}_1} = \frac{\partial \boldsymbol{f}}{\partial \boldsymbol{s}_1^\omega} \frac{\partial \boldsymbol{s}_1^\omega}{\partial \boldsymbol{u}_1} \text{ , and} \qquad\qquad \frac{\partial \boldsymbol{f}}{\partial \boldsymbol{u}_2} = \frac{\partial \boldsymbol{f}}{\partial \boldsymbol{s}_2^\omega} \frac{\partial \boldsymbol{s}_2^\omega}{\partial \boldsymbol{u}_2}$$

where the partial derivatives are given by basis functions $^1\phi$ and $^2\phi$ associated to $\boldsymbol{u}_1$ and $\boldsymbol{u}_2$:

$$\frac{\partial \boldsymbol{s}_1^\omega(c',c,x,y)}{\partial(\boldsymbol{u}_1)_j} = {}^1\phi_j^{c',c}(x,y) \text{ , and} \qquad\qquad \frac{\partial \boldsymbol{s}_2^\omega(c',c,x,y)}{\partial(\boldsymbol{u}_1)_j} = {}^2\phi_j^{c',c}(x,y)$$

As basis functions form partial derivatives from $\boldsymbol{s}_1^\omega$ and $\boldsymbol{s}_2^\omega$ to $\boldsymbol{u}_1$ and $\boldsymbol{u}_2$, respectively, we can use them to project the factors used in Section 3.1 in terms of the new sparsified parameters.

For sparsified convolutional layers S-CONV, we extend the KFAC approximation for convolutional layers derived in [Grosse and Martens, 2016]. Similarly to above, we use the existing derivation to approximate KFAC in terms of Jacobians w.r.t. induced convolutional filters $\boldsymbol{J}_{\bar{\boldsymbol{s}}^\omega}$. Filters $\bar{\boldsymbol{s}}^\omega$ in S-CONV layers are not part of the model parameters anymore and we need to find Jacobians $\boldsymbol{J}_{\boldsymbol{u}}$ with respect to anchor point weights $\bar{\boldsymbol{u}}$, which we can find by applying the chain rule:

$$\frac{\partial \boldsymbol{f}}{\partial \boldsymbol{u}} = \frac{\partial \boldsymbol{f}}{\partial \bar{\boldsymbol{s}}^\omega} \frac{\partial \boldsymbol{s}_2^\omega}{\partial \boldsymbol{u}_2}, \text{ with} \qquad\qquad \frac{\partial \bar{\boldsymbol{s}}^\omega(c',c,u,v)}{\partial \bar{\boldsymbol{u}}_j} = \bar{\phi}_j^{c',c}(u,v)$$

Similar to S-FC layers, we can project Kronecker factors of KFAC for CONV layer into the right sparser parameter space by multiplying the appropriate dimensions with basis function evaluations.

# D Inspecting learned layer-wise structure.

| | Prior precisions ↑ implies 'off' | | Weight norms ↓ implies 'off' | | Normalised Effective Num. of Param. ↓ implies 'off' | | Relative Effective Num. of Param. ↓ implies 'off' | | Effective |
|---|---|---|---|---|---|---|---|---|---|
| Layer $l$ | FC $\sigma_l^{-2}$ | CONV $\bar{\sigma}_l^{-2}$ | FC $\|\theta_l\|_2^2$ | CONV $\|\bar{\theta}_l\|_2^2$ | FC $\frac{\gamma_l}{\gamma_l+\bar{\gamma}_l}$ | CONV $\frac{\bar{\gamma}_l}{\gamma_l+\bar{\gamma}_l}$ | FC $\frac{\gamma_l/P_l}{\gamma_l/P_l+\bar{\gamma}_l/\bar{P}_l}$ | CONV $\frac{\bar{\gamma}_l/\bar{P}_l}{\gamma_l/P_l+\bar{\gamma}_l/\bar{P}_l}$ | Layer type |
| 0 | 3588348.50000000 | 0.07513507 | 0.00000000 | 10.74257360 | 0.00000002 | 0.80731778 | 0.00000003 | 0.99999997 | CONV |
| 1 | 2750749.50000000 | 1.26366782 | 0.00000000 | 0.55490241 | 0.00000007 | 0.70107951 | 0.00000010 | 0.99999990 | CONV |
| 2 | 5898884.50000000 | 14.50690365 | 0.00000000 | 0.04562404 | 0.00000014 | 0.66181451 | 0.00000021 | 0.99999979 | CONV |
| 3 | 22690960.00000000 | 230.75096130 | 0.00000000 | 0.00216724 | 0.00000089 | 0.50004517 | 0.00000178 | 0.99999822 | CONV |
| 4 | 29633508.00000000 | 5438.10449219 | 0.00000000 | 0.00007521 | 0.00000379 | 0.40905216 | 0.00001888 | 0.99998112 | CONV |
| 5 | 35631228.00000000 | 3797.48779297 | 0.00000000 | 0.00003772 | 0.00000379 | 0.14326709 | 0.00002643 | 0.99997357 | CONV |
| 6 | 635.57513428 | 31321916.00000000 | 0.00001135 | 0.00000000 | 0.00721232 | 0.00000522 | 0.99927627 | 0.00072373 | FC |
| 7 | 35582056.00000000 | 1.75766361 | 0.00000000 | 0.00259477 | 0.00000009 | 0.00456014 | 0.00001874 | 0.99998126 | CONV |

Table 4: Analysis of learned layer types after training FC+CONV model on CIFAR-10. Reported prior precision, weight norms, effective dimension and relative effective dimension metrics. On this task, the model learns to use CONV layers, except for a FC layer at the end of the model.

## D.1 Learned prior variances

To learn symmetries, we follow Bayesian model selection by specifying symmetry in the prior and learning it by optimising marginal likelihood estimates. For each layer, we consider a Gaussian prior with prior precisions $\frac{1}{\sigma^2}$ controlling the amount of equivariance. Most notably, in the limit of $\lim_{c\to\infty}\frac{1}{\sigma^2}=c$ we have strict equivariance. We can inspect prior precisions of both the CONV path $\bar{\sigma}^{-2}$ and the FC path $\sigma^{-2}$, where high prior precisions can intuitively interpret as the layer being 'switched off'. Prior precisions learned for the CONV+FC model are shown on the left in Table 4. Although prior precisions may not always be directly interpretable (see App. D.2, the learned prior precisions indicate that the model learns convolutional structure for most layers, except for layer 7 at the end, which we argue is a desirable solution as hypothesised in Section 6.2.

## D.2 Effective number of parameters

MacKay [1992] defines the effective number of parameters $\gamma$ using the Laplace approximation to the posterior. In particular, it measures how well a parameter is determined relative to its prior. MacKay [1992] defines it for a model with Gaussian prior $\mathcal{N}(0, \alpha^{-1}\boldsymbol{I}_P)$ over $P$, whose posterior covariance is given by $\boldsymbol{\Sigma}$. We then have

$$\gamma = P - \alpha \operatorname{Tr}(\boldsymbol{\Sigma}) = P - \alpha \operatorname{Tr}(\boldsymbol{H} + \alpha\boldsymbol{I}_P)^{-1} = P - \sum_{p=1}^{P}\frac{\alpha}{\lambda_p+\alpha} = \sum_{p=1}^{P}\frac{\lambda_p}{\lambda_p+\alpha},$$

where $\boldsymbol{H}$ is the Hessian at the posterior mode. However, this therefore requires the full posterior covariance $\boldsymbol{\Sigma}$ due to the Laplace approximation. In our case, we further have a layer-wise KFAC approximation over the individual layers $l$ with corresponding parameters. Our models share the following setup: we have a prior variances $\sigma_l^2$ and $\bar{\sigma}_l^2$ per layer, each corresponding to the fully connected and convolutional path, respectively. Further, let $P_l$ and $\bar{P}_l$ be the number of distinct parameters in the respective layers. Our posterior approximation is block-diagonal and Kronecker-factored, i.e., we have that $\boldsymbol{H}$ is a block-diagonal constructed from $\boldsymbol{H}_l$ and $\bar{\boldsymbol{H}}_l$ for all layers $l$ and each can be written as $\boldsymbol{H}_l \approx \boldsymbol{Q}_l \otimes \boldsymbol{W}_l$. Therefore, we have

$$\gamma = \sum_{l=1}^{L} P_l - \sigma_l^{-2}\operatorname{Tr}(\boldsymbol{H}_l + \sigma_l^{-2})^{-1} + \bar{P}_l - \bar{\sigma}_l^{-2}\operatorname{Tr}(\bar{\boldsymbol{H}}_l + \bar{\sigma}_l^{-2})^{-1}$$

$$= \sum_{l=1}^{L} \underbrace{\sum_{w,q}\frac{\lambda_{l,w}\lambda_{l,q}}{\lambda_{l,w}\lambda_{l,q}+\sigma_l^{-2}}}_{=\gamma_l} + \underbrace{\sum_{w,q}\frac{\bar{\lambda}_{l,w}\bar{\lambda}_{l,q}}{\bar{\lambda}_{l,w}\bar{\lambda}_{l,q}+\bar{\sigma}_l^{-2}}}_{=\bar{\gamma}_l},$$

where $\lambda_{l,w}$ denotes the eigenvalue of $\boldsymbol{W}_l$ and similarly for $\lambda_{l,q}$. The simplification is due to the eigenvalues of the Kronecker product being the outer product of its factors eigenvalues. Further, denote $\gamma_l$ and $\bar{\gamma}_l$ the effective number of parameters for FC and CONV layer at $l$. For FC and CONV, respectively, we define the 'normalised effective number of parameters' as $\frac{\gamma_l}{P_l}$ and $\frac{\bar{\gamma}_l}{\bar{P}_l}$ and the 'relative effective number of parameters' as $\frac{\gamma_l/P_l}{\gamma_l/P_l+\bar{\gamma}_l/\bar{P}_l}$ and $\frac{\bar{\gamma}_l/\bar{P}_l}{\gamma_l/P_l+\bar{\gamma}_l/\bar{P}_l}$. The definitions generalise trivially if more than these two layer types are being considered.

# E   Comparison of existing approaches

| Method | Symmetry learning | Layer-wise equivariance | Automatic objective | No validation data | No explicit regulariser | Sparse layer-wise symmetry |
|---|---|---|---|---|---|---|
| Zhou et al. [2019] | ✓ | ✓ | ✓ | ✓ | ✓ | |
| Benton et al. [2020] | ✓ | | | ✓ | | |
| Zhou et al. [2020] | ✓ | ✓ | ✓ | | ✓ | |
| Romero and Lohit [2021] | ✓ | ✓ | | | ✓ | |
| Finzi et al. [2021a] | ✓ | ✓ | | ✓ | | |
| Dehmamy et al. [2021] | ✓ | ✓ | | ✓ | ✓ | ✓ |
| Immer et al. [2022] | ✓ | | ✓ | ✓ | ✓ | |
| Yeh et al. [2022] | ✓ | ✓ | ✓ | | ✓ | |
| Maile et al. [2022] | ✓ | ✓ | ✓ | | ✓ | |
| van der Ouderaa et al. [2022] | ✓ | ✓ | | ✓ | | |
| Yang et al. [2023] | ✓ | ✓ | | ✓ | | |
| van der Ouderaa and van der Wilk [2023] | ✓ | ✓ | | ✓ | | ✓ |
| (**Ours**) | ✓ | ✓ | ✓ | ✓ | ✓ | ✓ |

Table 5: Overview of existing methods for symmetry discovery.

**Symmetry learning:**    In this comparison, we include methods that learn symmetries in the context of deep neural networks, including both learnable invariances and equivariances.

**Layer-wise equivariance:**    Invariances can often be parameterised by averaging data augmentations after forward passes through a model. Layer-wise equivariances can be more difficult to parameterise, as this places constraints on intermediary features and, therefore, relies on adaptions inside the actual architecture of the model.

**Automatic objective:**    Symmetries enforce constraints on the functions a network can represent, which makes it hard to learn them with regular maximum likelihood training objectives that rely on data fit. To avoid collapse into solutions with the least (symmetry) constraints, some methods rely on explicit regularisation to enforce symmetry. Although this approach has been successful in some cases [Benton et al., 2020, Finzi et al., 2021a, van der Ouderaa et al., 2022], issues with the approach have also been noted and discussed in Immer et al. [2022]. The most important critique is that direct regularisation of symmetry often depends on the chosen parameterisation and introduces additional hyperparameters that need tuning. Automatic objectives learn symmetries from training data without introducing additional hyperparameters that need tuning.

**No validation data:**    No validation data is used to select or learn symmetries. Most notably, these are methods that use differentiable validation data as an objective to learn hyperparameters. We also include methods in this category that initialise networks at strict symmetry and use validation data as an early-stopping criterion, in case no other encouragement is added in the training objective - assuming that early-stopping, in this case, becomes the mechanism that prevents collapse into non-symmetric solutions.

**Sparse layer-wise symmetry:**    We deem methods that fall within a 10-fold increase of number of parameters to parameterise relaxed symmetry constraints. Although this might seem like a lot, this distinguishes methods that relax symmetry by considering fully-flexible linear maps, which in practice often leads to an increase of more than 100 times.

# F  Implementation details

## F.1  Network architecture

Table 6 describes the used architecture for all experiments. The design is adapted from the convolutional architecture used in [Neyshabur, 2020]. We use strict equivariant subsampling described in App. F.2 as POOL and use element-wise ReLU$(x) = \max(0, x)$ activation functions. With convolutional layers CONV, the architecture is strictly equivariant. In some experiments the CONV layers are replaced to relax strict layer-wise symmetry constraints.

| layer | input | output |
|---|---|---|
| CONV($3 \times 3$) + ReLU | $B \times C_{\mathrm{in}} \times 32 \times 32$ | $B \times \alpha \times 32 \times 32$ |
| CONV($3 \times 3$) + POOL + ReLU | $B \times \alpha \times 32 \times 32$ | $B \times 2\alpha \times 16 \times 16$ |
| CONV($3 \times 3$) + ReLU | $B \times \alpha \times 16 \times 16$ | $B \times 2\alpha \times 16 \times 16$ |
| CONV($3 \times 3$) + POOL + ReLU | $B \times 2\alpha \times 16 \times 16$ | $B \times 4\alpha \times 8 \times 8$ |
| CONV($3 \times 3$) + ReLU | $B \times 4\alpha \times 16 \times 16$ | $B \times 4\alpha \times 8 \times 8$ |
| CONV($3 \times 3$) + POOL + ReLU | $B \times 4\alpha \times 8 \times 8$ | $B \times 8\alpha \times 4 \times 4$ |
| CONV($3 \times 3$) + POOL + ReLU | $B \times 8\alpha \times 4 \times 4$ | $B \times 8\alpha \times 2 \times 2$ |
| CONV($3 \times 3$) + ReLU | $B \times 16\alpha \times 2 \times 2$ | $B \times 16\alpha \times 1 \times 1$ |
| CONV($1 \times 1$) + ReLU | $B \times 16\alpha \times 1 \times 1$ | $B \times 64\alpha \times 1 \times 1$ |
| CONV($1 \times 1$) | $B \times 64\alpha \times 1 \times 1$ | $B \times C_{\mathrm{out}} \times 1 \times 1$ |

Table 6: Used architecture with $\alpha = 10$.

## F.2  Group subsampling and maintaining strict equivairance.

The core idea of this work is to utilise parameterisations that have strict equivariance symmetries as controllable limiting case and learning the relative importance of the symmetry constraint through Bayesian model selection. In this framework, we encode the limiting case of strict equivariance in the prior through use of (group) convolutions with circular padding and pointwise non-linearities, which respect the equivariance property. It is known, however, that subsampling operations such as maxpooling and strided convolutions present in most convolutional architectures, including our baseline [Neyshabur, 2020], do not respect equivairance strictly. To overcome this issue, we might consider coset-pooling [Cohen and Welling, 2016], but this looses locality of the feature maps and is therefore typically only used at the end of the architecture. Alternatives, such as Xu et al. [2021], introduce an additional subsampling dimension which requires memory and complicates implementation. The implementation of our method is already involved because Kronecker-factored curvature approximations are to date not naively supported in most deep learning frameworks. We therefore follow the more pragmatic approach of Chaman and Dokmanic [2021] and subsample feature maps by breaking them up into polyphase components selecting the component with the highest norm (we use $l_\infty$-norm). This is easy to implement and maintains both strict equivairance and locality of feature maps. Furthermore, the baseline is not negatively affected by this change as regular MAP performance of the CONV model improved by 1-2 % percentage points in test accuracy.

## F.3  Locality of support, separable group convolutions, pointwise group convolutions.

In deep learning literature, convolutional filters commonly only have a very small local support (e.g. $5 \times 5$, $3 \times 3$ or even $1 \times 1$). This is efficient and induces a locality bias that has empirically been found beneficial for many tasks. Interestingly, group equivariant convolutional filters are rarely locally supported and often defined on the entire group. Recent work by Knigge et al. [2022], explores factoring filters by generalising the concept of (spatially) separable convolutions to (sub)groups. Spatial separability implies the filter factorisation $\bar{\boldsymbol{\theta}}(c', c, x, y) = \bar{\boldsymbol{\theta}}(c', c, x)\bar{\boldsymbol{\theta}}(c', c, y)$. Applying the same concept to separate rotation and translation subgroups for roto-translation equivariance with $(x, y, \theta) \in G = \mathbb{Z}^2 \rtimes p4$ factors filters over subgroups $\bar{\boldsymbol{\theta}}(c', c, x, y, \theta) = \bar{\boldsymbol{\theta}}(c', c, x, y)\bar{\boldsymbol{\theta}}(c', c, \theta)$. We take this one step further and propose filters that are only pointwise for specific subgroups, which can be seen as a special case of separable group convolution with $\bar{\boldsymbol{\theta}}(c', c, \theta) = \mathbf{1}_{\theta = \mathrm{Id}}(\theta)$ with the indicator function always returning 1 at the group identity element Id $\in G$ and 0 otherwise. Pointwise group convolutions generalise commonly used $1 \times 1$ convolutional filters to (sub)groups. The simplification does not impact the equivairance property (to both rotation and translation) in any way.

Pointwise group convolutions are easier to implement because the summation over the (sub)group filter domain can be omitted, reducing memory and computational cost in the forward-pass by a factor proportional to the (sub)group size $|G|$. In Table 7, we compare full, separable and pointwise factorisations of the rotation subgroup in a roto-translation equivariant CNN. We also note an interesting equivalence between applying group convolutions that are pointwise in specific (sub)groups and invariance (to specific subgroups) that is achieved by pooling outputs after applying the same network to augmented inputs on the group. This connects invariance obtained by pooling after stacked layer-wise equivariant layers with DeepSets [Zaheer et al., 2017] and the set-up used in most invariance learning literature [Benton et al., 2020, van der Ouderaa and van der Wilk, 2021, Immer et al., 2022]. The equivalence only holds for strict equivariance. Pointwise convolutions have the benefit that they can be relaxed using our method and allow place-coded features related to the group to rate-coded features, on a per-layer basis.

| Roto-translation ($Z^2 \rtimes p4$)-CNN | $\bar{\theta}(c', c, x, y, \theta)$ | Parameters | $(\alpha)$ | Rotated CIFAR-10 Test accuracy (%) |
|---|---|---|---|---|
| Full, channel matched | $\bar{\theta}(c', c, x, y, \theta)$ | 1025380 | (10) | 73.70 |
| Separable, channel matched | $\bar{\theta}_1(c', c, x, y)\bar{\theta}_2(c', c, \theta)$ | 214550 | (10) | 61.31 |
| Depthwise Separable, channel matched | $\bar{\theta}_1(x, y)\bar{\theta}_2(\theta)\bar{\theta}_3(c', c)$ | 139735 | (10) | 51.63 |
| Full, parameter matched | $\bar{\theta}(c', c, x, y, \theta)$ | 371200 | ( 6) | 69.89 |
| Separable, parameter matched | $\bar{\theta}_1(c', c, x, y)\bar{\theta}_2(c', c, \theta)$ | 358463 | (13) | 59.99 |
| Depthwise Separable, parameter matched | $\bar{\theta}_1(x, y)\bar{\theta}_2(\theta)\bar{\theta}_3(c', c)$ | 346215 | (16) | 49.77 |
| Pointwise, channel+parameter matched | $\bar{\theta}_1(c', c, x, y)\mathbf{1}_{\theta=\mathrm{Id}}(\theta)$ | 338770 | (10) | 66.77 |
| Classical CNN, baseline | $\bar{\theta}(c', c, x, y)$ ($\theta$ not convolved) | 338770 | (10) | 57.89 |

Table 7: Experimental comparison of roto-translation equivariant convolutional architectures with full, separable and pointwise factorisation of the rotation subgroup. Reported test accuracy.

### F.4   Sparsified layers

For sparsified S-FC and S-CONV layers, we use half the number of anchor points $M$ as there would otherwise be parameters in that layer without spatial sparsification. In non-integer, we round upwards.

### F.5   Datasets

For MNIST [LeCun, 1998] and CIFAR-10 [Krizhevsky et al., 2009] datasets, we normalise images standardised to zero mean and unit variance, following standard practice. Rotated datasets 'Rotated MNIST' and 'Rotated CIFAR-10' consists of original datasets but every image rotated by an angle uniformly sampled from along the unit circle. Similarly, images in 'Translated MNIST' are randomly translated in x- and y- axes by uniformly sampled pixels in the range [-8, 8].

### F.6   Training details

For the CIFAR-10 experiments, we optimise using Adam [Kingma and Ba, 2014] ($\beta_1$=0.9, $\beta_2$=0.999) with a learning rate of 0.01 for model parameters $\boldsymbol{\theta}$ and 0.1 for hyperparameters $\boldsymbol{\eta}$, cosine-annealed [Loshchilov and Hutter, 2016] to zero. We train with a batch size of 128 for 4000 epochs, and update hyperparameters every 5 epochs after a 10 epoch burn-in. For parameter updates, we use standard data augmentation consisting of horizontal flips and 4-pixel random shifts, and do not augment when calculating the marginal likelihood estimates. For MAP training, we use unit prior variances $\sigma_l = \bar{\sigma}_l = 1$ for all layers $l$. For Laplace training, we use this same setting to initialise our prior but treat them as part of the hyperparameters $\boldsymbol{\eta}$ and optimise them during training. For MNIST experiments, we use the same settings, except that we do not use data augmentation and train for a shorter period of 1000 epochs at a lower initial learning rate of 0.01. All experiments were run on a single NVIDIA RTX 3090 GPU with 24GB onboard memory.

### F.7   Code

Code accompanying this paper is available at `https://github.com/tychovdo/ella`