# OpenReview forum: "Learning Layer-wise Equivariances Automatically using Gradients"
_NeurIPS.cc/2023/Conference — NeurIPS 2023 spotlight_

### Official Review · Reviewer_n5aj · 2023-06-19

**Soundness:** 3 good
**Presentation:** 2 fair
**Contribution:** 2 fair
**Rating:** 7
**Confidence:** 3

**Summary:**

Edit: Rating updated from 6 to 7 after rebuttal.

The goal of the paper is to learn an interpolation between non-equivariant and equivariant models. The authors introduce different convolutional and non-convolutional linear layers, optionally being sparsified via factorizations or a smooth spatial basis. The basic idea is to define the model as a linear combination of translation-equivariant and non-equivariant layers, and to optimize their relative contribution in order to select whether the final model is equivariant or not. This is achieved by posing different Gaussian priors on their parameters, whose widths constitute hyperparameters to be optimized via Bayesian model selection. This model selection differs from prior work, which tuned the hyperparameters explicitly.

Bayesian model selection requires the evaluation of a marginal likelihood term, which is infeasible. This is addressed via the Laplace approximation of the parameter posterior proposed by Immer et al. (2022). As this would still require the computation of a Hessian matrix which scales quadratically in the number of parameters, a Kronecker factorization approximation is used.

Experiments on image classification datasets show that the interpolated models outperform non-equivariant models and, on tasks with broken equivariance, also strictly equivariant models. The latter are, in turn, performing slightly better on strictly equivariant learning tasks. An ablation to a pure MAP optimization baseline shows that the model selection approach results in significant gains.
Results for further groups beyond translations are discussed in the appendix and briefly evaluated in the experimental section.

**Strengths:**

The paper combines ideas from prior work and demonstrates a superior performance compared to these baselines. It relies on a Bayesian approach and shows how it can be made feasible via approximations despite the analytical intractability and high dimensionalities of parameter spaces. Automatically learning hyperparameters is certainly an improvement over manual tuning. The presented empirical evidence shows that the model selection does indeed select equivariant or non-equivariant layers depending on the symmetries of the learning task.

**Weaknesses:**

The main downside of the approach is that it does not really learn equivariant models from scratch, but rather learns to select between pre-specified models with different levels of equivariance. However, this downside is shared by a line of prior work, which the current approach improves upon.

I am also not entirely convinced that learning such selections is practically relevant since the appropriate levels of equivariance groups are usually known a-priori or become evident when comparing an equivariant model against a non-equivariant baseline. The experiments suggest that using strictly equivariant models is working better in cases where the desired equivariance group is indeed known a-priori.

Instead of describing general equivariant mappings, the main paper considers only conventional convolutions, i.e. translation equivariance. More general results are discussed in the supplementary material and briefly evaluated in the last experimental section. It would have been nice to have a more general formulation in the main paper.

It would also have been interesting to see how the method scales to multiple symmetry groups at once. For instance, one could consider all 2^3 (i.e. exponentially many) combinations of using translations, rotations and reflections.

The explanation of the method could also be more clear. I had to re-read section 3 and the exact nature of the hyperparameters, introduced in section 3, remained vague till section 4.2. It would have been easier to just mention model selection of hyperparameters in the intro and moving the current section 3 between sections 4 and 5.

**Questions:**

The authors state that "adding symmetries does not directly improve training losses that rely on training fit", however, many papers on equivariant models show their improved convergence rate and final loss when being fitted. Could this statement be clarified and supported more explicitly with evidence?

The notation of the limit in equation 9 does not seem to make sense. I guess that the authors just want to say that the implication follows given $\sigma^2=0$?
I found the notation of the feature map's domain as $\mathbb{Z}^3$ somewhat confusing and would rather write $[0,C]\times \mathbb{Z}^2$ (spatially supported on $[0,X]\times[0,Y]$).
More questions/suggestions are found in the "weakness" section.

**Limitations:**

Limitations are not explicitly discussed. I do not have ethical concerns

---

> ### Author Rebuttal · Authors · 2023-08-09
>
> Thank you for your feedback and help to improve the paper.
>
> > learns to select between pre-specified models with different levels of equivariance. However, this downside is shared by a line of prior work + scales to multiple symmetry groups at once
>
> The approach is general in that it can work with general groups, as long as they can be differentiably parameterised. Learning symmetries using training data only from simple affine transformations is still actively investigated [1, 2, 3], and we extend the parameterisations to allow for layer-wise equivariances and provide concrete improvements to the scalability of parameterisations. Parameterising distributions over other groups and scaling to multiple symmetry groups is an interesting avenue for future work, but orthogonal to enabling differentiably learning of such parameterisations, which is what our paper focuses on.
>
> > using strictly equivariant models is working better in cases where the desired equivariance group is indeed known
>
> We agree with the reviewer that when a strict global symmetry is known a-priori or self-evident, it would always be better just to build it into the model. However, in some cases, the symmetry might not be known, specified, or should not be strictly enforced (think 6’s and 9’s in MNIST under rotational symmetry). Finding the right symmetry or optimal balance between equivariant and non-equivariant layers becomes harder on more complex models and datasets, especially when this can also differ layer-wise.
>
> On image classification tasks where equivariance seems very desirable, the standard MAP objective cannot select and ignore the non-equivariant pathways. Yet, our proposed Diff Laplace objective based on approximate Bayesian model selection does allow symmetry learning and can automatically select the most relevant symmetry, becoming largely equivariant on this task. This demonstrates the ability to learn relevant layer-wise symmetries from training data automatically.
>
> >discussed in the supplementary material and briefly evaluated in the last experimental section
>
> We will follow the suggestion by the reviewer to move the description of general group equivariant mappings to the main text.
>
> > moving the current section 3 between sections 4 and 5.
>
> We agree with the reviewer and will adopt the suggestion to move Sec. 3 between Sec. 4 and Sec. 5 to make the paper clearer.
>
> > The authors state that "adding symmetries does not directly improve training losses that rely on training fit", however, many papers on equivariant models show their improved convergence rate and final loss when being fitted. Could this statement be clarified and supported more explicitly with evidence?
>
> There is significant evidence that MAP alone cannot learn hyperparameters, including symmetry constraints. This is supported by experiments performed in earlier work on invariance/equivariance learning [1,2,3], with explicit examples that MAP cannot learn symmetry constraints in Sec. 4.2 of [2] and App. C of [3]. Because of this reason, works in literature often require differentiated validation losses [5, 6], explicit regularisation [7], or RL outer loops [4]. We hope this shows that problems with the MAP objective are well-established facts.
>
> The reason that such a data-fitting objective does not encourage symmetry is that symmetries constrain the functions that a network can represent. Consequently, the regular objective that maximises train data fit will always prefer as little as possible equivariance (no symmetry, so σ=0), even when more equivariance would be preferable in terms of better generalisation on validation/test data.
> In some particular cases, as the reviewer mentions, it can happen that there is an improved training fit when using the right equivariance. However, this only occurs when model capacity is constrained. To properly compare two architectures, they need to be large enough for their performance not to be artificially restrained by a lack of size. This is where the difference in training fit disappears and where an additional measure of complexity needs to be used, which the Laplace approximation provides.
>
> We have seen similar claims about faster convergence rates in models with the right equivariance, and we share the reviewer's view that this is a phenomenon that _may_ allow point optimisation to favour architectures that generalise well. However, there are currently no papers that successfully demonstrate using convergence rates for architecture selection, and our attempts to use this have failed, similarly to [1,2,3].
>
> We show that by using the proposed Diff Laplace objective using approximate Bayesian model selection, we can in fact automatically learn symmetries using training data.
>
> > notation of the limit in equation 9
>
> Indeed. Eq.9 should be the implication that follows given $\sigma^2 = 0$. We will fix this.
>
> > I found the notation of the feature map's domain as somewhat confusing
>
> We thank the reviewer for pointing this out and adopting the suggested change in notation.
>
>
> [1] Wilk, Mark van der, et al. "Learning invariances using the marginal likelihood." NeurIPS 2018
>
> [2] van der Ouderaa, Tycho FA, et al. "Learning invariant weights in neural networks." UAI 2022
>
> [3] Immer, Alexander, et al. "Invariance learning in deep neural networks with differentiable Laplace approximations." NeurIPS 2022
>
> [4] Cubuk, Ekin D., et al. "Autoaugment: Learning augmentation strategies from data." CVPR 2019
>
> [5] Lorraine, Jonathan, et al. "Optimizing millions of hyperparameters by implicit differentiation." AISTATS 2020
>
> [6] Liu, Hanxiao, et al. "Darts: Differentiable architecture search." arXiv preprint arXiv:1806.09055 (2018).
>
> [7] Benton, Gregory, et al. "Learning invariances in neural networks from training data." NeurIPS 2020
>
> Typos will be fixed.

---

> > ### Comment · Reviewer_n5aj · 2023-08-11
> >
> > I would like to thank the authors for their detailed reply. My concerns are adequately addressed and I am happy with the promised updates in the paper. I updated my rating from 6 to 7.

---

### Official Review · Reviewer_VChg · 2023-07-05

**Soundness:** 3 good
**Presentation:** 3 good
**Contribution:** 2 fair
**Rating:** 7
**Confidence:** 4

**Summary:**

While (group) convolutions encode strict symmetries into neural network architectures, this paper presents a method for representing flexible symmetry constraints and learning the degree of symmetry automatically (through marginal likelihood objectives). Their method builds on residual pathways to represent each NN layer as the sum of fully connected and convolutional layers, where the initialization variance hyperparameter of the FC weights controls the degree of equivariance and is automatically optimized through the marginal likelihood. They introduce a number of techniques to make this process tractable in practice:
* The FC layer is factored along spatial dimensions and possibly sparsified
* To efficiently compute a laplace approximation of the marginal likelihood, they use a KFAC approximation of the Hessian. The factorization/sparsification of the FC layers admit additional simplifications to the KFAC Hessian computation.

The paper includes experiments comparing this method with plain FC or CNN architectures on synthetic and natural image datasets (MNIST and CIFAR-10), and show that it can learn to adjust the degree of equivariance to achieve good performance on these tasks. Analyzing the optimized hyperparameters shows that the method prefers to make earlier layers more equivariant and later layers less equivariant, in agreement with common architecture design principles. Finally, they show the ability to select between multiple symmetry groups (Conv and 90degree rotation GConv) on CIFAR10.

**Strengths:**

The paper introduces a number of technical innovations to tackle a very challenging problem, automatic symmetry discovery. Marginal likelihood optimization removes the need for validation datasets or handcrafted regularizers when learning symmetries (which were used by some prior work), but is difficult to compute efficiently at the scale of modern NNs. It also attempts to address a common problem of symmetry discovery work where, if you start with no symmetry assumptions (FC), you end up with an enormous number of parameters for high dimensional inputs which is not computationally feasible in practice. The authors show that factoring (along spatial dimensions) and sparsifying the FC layers can reduce the number of parameters and even admit more efficient marginal likelihood estimation.

**Weaknesses:**

The empirical results are limited and do not demonstrated that this method will be broadly applicable, in my opinion. The real datasets considered (CIFAR-10 and MNIST) are relatively simple and existing techniques (e.g., strong data augmentation or SSL + resnets or ViT) likely obtain much stronger results on these benchmarks without needing symmetry learning. Although the purported advantage of symmetry learning is that it can hopefully do better than human chosen symmetry constraints (like humans choosing data augmentations), the empirical results here don't show that.

Although in principle the method is applicable to any symmetry group, the experiments seem to focus almost exclusively on translation invariance. Only 6.3 studies the case with 2 possible symmetries: translation (conv) and rotation (GConv). Is there a concern with scalability to more possible symmetries? Ideally, we would like the method to be able to learn any relaxed symmetries more generally without having to restrict the search space to one or two options (translation and rotation).

Relatedly, much of the paper and in particular the spatial factorization seem particularly suited to image inputs (or similar input modalities). Would spatial factorization still work well with other data modalities, like graphs representing molecules? The experiments also seem to be focused on image inputs only.

Related work: the experiments and aims of this paper are reminiscent to [Elsayed et al, 2020](https://arxiv.org/abs/2002.02959), which also aimed to learn related spatial symmetry by operating on a spectrum between locally connected and convolutional layers. Using locally connected layers can also be viewed as "sparsifying" the fully connected layer to decrease computational costs. It would be interesting to discuss the differences and similarities here, either in text or in terms of empirical results (or both).

**Questions:**

Some minor things:

What is P in Eq 1, the number of parameters? Might've missed this.

Why does the definition of fully connected preserve the spatial dimensions X and Y? In principle, I'd expect that a fully connected layer (or a conv layer) can change the spatial dimensions. Also, the closing bracket $[0,X$ is missing above Eq 2.

Section 6.2: If I understand properly, isn't the optimal behavior just to ignore the fully connected pathway altogether and set its variance -> 0?

[Post-rebuttal update]: I have read the author's response. They answer most of my questions. I largely maintain my original score and evaluation.

**Limitations:**

The authors discuss limitations of their method and the general line of work.

---

> ### Author Rebuttal · Authors · 2023-08-09
>
> Thank you for your feedback and help to improve the paper.
>
> > empirical results are limited
>
> Even when strict symmetries are desirable, our method shows that such favourable architectures can be discovered automatically, whereas the standard MAP objective cannot. Our proposed method allows automatically learning layer-wise equivariances from training data through approximate Bayesian model selection. We have demonstrated this principle but agree with the reviewers that assessing performance on larger real-world datasets and models would be interesting.
>
> > method is applicable to any symmetry group, but experiments focus on translation invariance
>
> We describe the method for general groups and demonstrate symmetry selection between multiple groups in Sec. 6.3. Although some prior works have considered a more extensive range of groups (e.g. [1]), we have a greater focus on practical scalability of the parameterisations (Sec. 4) and puts a considerable amount of work in the objective function (Sec. 3, 5). In doing so, we have demonstrated the principle of automatically selecting the most relevant layer-wise equivariances from training data through Bayesian model selection. We agree that it would be interesting to consider more groups in future work.
>
> > scalability to more possible symmetries?
>
> Scalability to more symmetries likely depends on the exact problem set-up and considered groups and representations. Parameterising group symmetries is a research field in its own right and not the focus of this work. Nevertheless, the existing parameterisation of neural networks with many symmetries and regular group representations should be readily extendable to relaxed and learnable symmetries using the techniques presented in this paper. We see no direct concern with scalability to more symmetries. This is an interesting avenue for future research, but orthogonal to enabling learnable layer-wise symmetries from training data, which this work focuses on.
>
> > Would spatial factorisation still work well with other data modalities, like graphs representing molecules?
>
> Yes, the spatial factorisation should generalise to other data modalities as long as you can construct basis functions on the manifolds [2], which are also available for graphs [4].
>
> > Related work: the experiments and aims of this paper are reminiscent to Elsayed et al, 2020,
>
> We greatly thank the reviewer for pointing this out and will include [3] in the related work.
>
> There are indeed some interesting parallels between the work and the parameterisation aspects (Sec. 4) of our work. Indeed, the motivation behind including spatial dependence is similar as well as recognising that "combining weights scales linearly with the number of pixels in the image, which may be large." There seem to be subtle differences in how both methods address this. For instance, [3] uses 'low-rank locally connected layers' factorisation between height and width dimension, whereas our factorisation factors between the full input and output dimensions. Further, our sparsification uses basis features which do not necessarily result in a rank-1 matrix spatially, whereas the low-rank factorisation of [3] will.
>
> > What is P in Eq 1, the number of parameters?
>
> Indeed, P is the number of parameters. We will make this clear from the text.
>
> > Why does the definition of fully connected preserve the spatial dimensions X and Y?
>
> This indeed differs slightly from the typical formulation of fully-connected layers that allow different input and output dimensions. We do this mainly for notational simplicity and to avoid running into the complexity of having different group structures on the input and output. We are still absolutely flexible regarding changing spatial dimensions but only do this through equivariant pooling (see App. F). For instance, instead of defining an FC layer with 2x smaller output dimensions, we simply use an FC without spatial downsampling followed by a 2x2 filter spatial pooling layer. We go through all this effort to ensure that equivariant paths are strictly equivariant and that our experimental set-up can verifiably show that our method actually learns layer-wise symmetries. This might be less important when applying the method in practice.
>
> > If I understand properly, isn't the optimal behaviour just to ignore the fully connected pathway altogether and set its variance -> 0?
>
> Yes. Equivariance on the image classification task is likely desirable, thus ignoring the fully connected pathway. We refrain from making any hard claims about optimality but generally agree with the reviewer.
>
> In line with theory, we observe that this “optimal behavior” cannot be learned with the standard MAP objective but can be learned with Diff. Laplace. Using the proposed objective, the network `becomes equivariant’ and largely learns to ignore the fully connected pathways. This demonstrates the ability of our method to select the most relevant symmetry from training data automatically.
>
> [1] Finzi, Marc, Gregory Benton, and Andrew G. Wilson. "Residual pathway priors for soft equivariance constraints." NeurIPS 2021
>
> [2] Azangulov, Iskander, et al. "Stationary kernels and Gaussian processes on Lie groups and their homogeneous spaces i: the compact case." 2022
>
> [3] Elsayed, Gamaleldin, et al. "Revisiting spatial invariance with low-rank local connectivity." ICML 2020.
>
> [4] Borovitskiy, Viacheslav, et al. "Matérn Gaussian processes on graphs." AISTATS 2021.
>
> Typos will be fixed.

---

### Official Review · Reviewer_gE9j · 2023-07-07

**Soundness:** 4 excellent
**Presentation:** 4 excellent
**Contribution:** 3 good
**Rating:** 7
**Confidence:** 3

**Summary:**

This paper proposes a neural network architecture and gradient-based training algorithm for modeling approximately equivariant functions. The architecture builds upon residual pathway (Finzi et al., 2021), where each layer of network is parameterized as an additive combination of constrained equivariant path and unconstrained fully connected path. A general challenge for such architectures is that equivariance is not favored for fitting training data, so empirical loss minimization will likely result in unstructured solutions that do not generalize. While the residual pathway paper solves this by putting higher weight regularization on unconstrained path, the strength of regularization is a hyperparameter that requires search with validation data in principle. The main motivation in this paper is to learn the extent of equivariance in each layer only from training data in a single training run (online). For this, the authors adopt Bayesian model selection which allows gradient-based learning of hyperparameters from training data by maximizing marginal likelihood estimate, specifically chosen in this work to be Laplace approximation with structured Hessian approximation with KFAC. Given that, the major technical contributions of the paper are on (1) improving the parameter efficiency of residual pathway by introducing convolution on Lie groups as well as factorization and spatial sparsification based on standard exponential basis functions, and (2) specifying the extent of equivariance as hyperparameters controlling the priors placed on the parameters, and (3) deriving KFAC for the proposed parameterizations so that gradient-based learning of hyperparameters controlling the extent of equivariance is made possible through maximization of marginal likelihood estimate. The authors provide experiments mainly regarding discrete 2-dimensional translation symmetry, and demonstrate that the proposed algorithm can (1) learn partial symmetry when needed, (2) recover the standard architecture of convolutional stack postfixed by fully-connected layers, and (3) learn to select from multiple symmetry groups depending on task, solely from training data.

Finzi et al., Residual Pathway Priors for Soft Equivariance Constraints (2021)

**Strengths:**

S1. Overall, I think this is a solid work that contributes towards solving a challenging and important problem of learning symmetry from training data by bridging (approximate) equivariant architecture design and Bayesian deep learning. I find no critical issue with originality, quality, clarity, and significance of the work; the writing is overall clear, the design of the algorithm is clearly motivated and presented, and the experimental results seem to support the main claims and motivations.

**Weaknesses:**

W1. In Table 1-3, in addition to test performance, it would be nice if I could see how the models (over)fit to training data (i.e., how they generalize) given that a major motivation of the work comes from the fact that equivariance is not encouraged when fitting training data but beneficial for generalization.

W2. The algorithm is empirically demonstrated for discrete 2-dimensional translation and 90-degree rotation symmetries. While the authors argue that extension to more general groups is possible in principle, I find the empirical demonstration regarding the argument is limited compared to the residual pathway paper (Finzi et al., 2021) that provided comprehensive experiments regarding e.g., continuous orthogonal groups as well.

Finzi et al., Residual Pathway Priors for Soft Equivariance Constraints (2021)

**Questions:**

Q1. Considering the space of input and output feature maps, how is the group lifting (Appendix A) done in general when there is a set of M groups G1, …, GM (Appendix B)? Do you use direct product G = G1 x … x GM? This doesn’t seem straightforward, I might have missed something.

Q2. Does the derivation of KFAC in Appendix C extend to other groups in a similar way to Appendix A? It seems so intuitively, but I would like to ask for a clarification from the authors.

Q3. Do the layer indices 0-7 in Appendix D match each layer in Table 6? That is, can I regard the learned layer 6 in Appendix D to correspond to the layer that maps spatial dimension 4 x 4 to 2 x 2 in Table 6?

Q4. In Table 6, it is written that the architecture is used for all experiments, and CONV layers are marked with kernel size (e.g., 3 x 3). How should one interpret this for S-CONV layers where, to my understanding, kernel size is not fixed?

**Limitations:**

The authors have partially addressed limitations of the work in Section 7; I encourage further clarifications of limitations that needs to be addressed in future work, if there are any.

---

> ### Author Rebuttal · Authors · 2023-08-09
>
> Thank you for your feedback and help to improve the paper.
>
> > In Table 1-3, in addition to test performance, it would be nice if I could see how the models (over)fit to training data.
>
> Thank you for raising this. We agree with the author that providing training performance is important given the motivation of learning symmetries by balancing data fit and model complexity through approximate Bayesian model selection. We can confirm that all models fit the training data well, as measured by a low negative log likelihood. We will add performance on the train set to the final version.
>
> > the authors argue that extension to more general groups is possible in principle, I find the empirical demonstration regarding the argument is limited
>
> The method is described for general groups and an experiment demonstrating symmetry selection between multiple groups is given in Sec. 6.3. Although some prior works have considered a more extensive range of groups (e.g. [2]), our work has a much more extensive focus on the scalability of parameterisations and we put a considerable amount of work in deriving an objective function that can learn symmetry, which is a focus rather than how to parameterise to many groups. We demonstrate the principle of automatically selecting the most relevant layer-wise equivariances from training data through Bayesian model selection. We agree that it would be interesting to consider more groups in future work.
>
> > Q1. How is group lifting done in general?
>
> Parameterising neural networks to obey group symmetries is an entire research field in its own right. The choice of parameterisation of the symmetry and its representation may vary depending on task-specific constraints. The formalisation of group lifting can be found in [3]. When combining rotation and translation, we follow regular representations similar to [1]. This means that if the network learns to ignore non-equivariant paths entirely, it will become exactly equivalent to standard G-CNN (e.g. P4CNN in [1]).
>
> > Q2. Does the derivation of KFAC in Appendix C extend to other groups in a similar way to Appendix A?
>
> In our case, yes, since we define filters on a group through base features defined in a vector space [4] and can therefore readily be extended to KFAC without issues. KFAC only needs to properly handle the weight-sharing induced by the group parameterisation, which we explain how to do in Sec. 5. We do note, however, that parameterising symmetries in neural networks is a research field in its own right. Different parameterisations might affect how easily KFAC can be derived. We will add this disclaimer.
>
> > Q3. Do the layer indices 0-7 in Appendix D match each layer in Table 6?
>
> Yes. The same architecture is used for all experiments and layer numberings are consistent.
>
> > Q4. In Table 6, it is written that the architecture is used for all experiments, and CONV layers are marked with kernel size (e.g., 3 x 3). How should one interpret this for S-CONV layers where, to my understanding, kernel size is not fixed?
>
> The kernel size of an S-CONV layer is fixed and kept the same (e.g. a 3 x 3 filter) as the CONV layer. The difference between both layers is that the number of parameters is decoupled from the kernel size. That is, by using basis features we can build an S-CONV layer with a 3 x 3 filter with fewer (e.g. 4) parameters, instead of the 9 required by CONV. A more extensive explanation of sparsifying filters through basis features can be found in [4]. We will make this more clear from the main text.
>
>
> [1] Cohen, Taco, and Max Welling. "Group equivariant convolutional networks." International conference on machine learning. PMLR, 2016.
>
> [2] Finzi, Marc, Gregory Benton, and Andrew G. Wilson. "Residual pathway priors for soft equivariance constraints." Advances in Neural Information Processing Systems 34 (2021): 30037-30049.
>
> [3] Kondor, Risi, and Shubhendu Trivedi. "On the generalization of equivariance and convolution in neural networks to the action of compact groups." International Conference on Machine Learning. PMLR, 2018.
>
> [4] van der Ouderaa, Tycho FA, and Mark van der Wilk. "Sparse Convolutions on Lie Groups." NeurIPS Workshop on Symmetry and Geometry in Neural Representations. PMLR, 2023.

---

> > ### Comment · Reviewer_gE9j · 2023-08-18
> > **Response to rebuttal**
> >
> > Thank you for the response. I recommend the authors to include the discussion regarding W2, Q1, Q2 in the final version of the paper. I have no further questions for now.

---

### Official Review · Reviewer_iYmm · 2023-07-10

**Soundness:** 3 good
**Presentation:** 3 good
**Contribution:** 3 good
**Rating:** 6
**Confidence:** 3

**Summary:**

The work proposed an automatic way to learn equivariance in each layer by finding a balance between the equivariant layer and the unrestricted fully connected layer. Unlike the previous work on soft equivariance, the work proposed to learn the balance between them via Bayesian model selection using gradients. The work also proposes different parameter-reduction techniques and achieves better results in the conducted experiments.

**Strengths:**

1. The work provides a technique for learning equivariance structure automatically through gradient
2. The work addresses the overparameterization of such soft equivariant models by factorization and sparsifications.
3. The paper is well-organized and clearly written.

**Weaknesses:**

1. Evaluations: The empirical evaluations are conducted primarily on image data where the motivation for finding a balance between the equivariant and non-equivariant layers is not clear (except for the toy problem). Clearly, for image classifications, labels should not be affected by translation or rotation.  For the toy problem, a comparison with Finzi et al. [2021a] is not provided.
2. The performance of regular equivariant architecture is not provided, which makes it difficult to measure the gain by the proposed method.

Marc Finzi, Gregory Benton, and Andrew G Wilson. Residual pathway priors for soft equivariance
constraints. Advances in Neural Information Processing Systems

**Questions:**

1. I am curious about the memory and time (training) required for the proposed model compared to the regular equivariant architecture.
2. What may be the reason for MAP performing worse than the proposed Diff. Laplace method?

---

> ### Author Rebuttal · Authors · 2023-08-09
>
> We thank the reviewer for the feedback.
>
> > motivation for finding equivariance balance
>
> Symmetry discovery from data is a well-recognized task and is actively studied [1,2,5,6,7]. The key motivation is that the true symmetry may be unknown, not specified, or that the optimal balance of equivariant and non-equivariant layers is in fact not 'clear'. Examples of this already arise in simple tasks, such as 6’s and 9’s in MNIST or our translational dependence toy problem. Finding the right balance arguably becomes even more difficult on more complex data, architectures and when optimal symmetries differ per layer, which we focus on. As discussed in Sec. 6.2, common vision architectures deploy translational symmetries, but are often not strictly invariant globally due to fully connected layers at the end of the network. Our method paves the way to discovering such favourable architectures automatically from training data.
>
> Our work demonstrates the principle of automatically learning layer-wise equivariances through approximate Bayesian model selection for neural networks. Even where strict symmetry is most desirable, our method is able to ignore non-equivariant layers and `become equivariant’, whereas standard MAP objectives cannot.
>
> > performance of regular equivariant architecture is not provided
>
> All experiments include the performance of regular equivariant architectures denoted by 'CONV'.
>
> > memory and time (training)
>
> The proposed factorised F- and sparsified S- parameterisations improve both training time and memory easily by 2x at a negligible loss in performance. Tables 1-2 include the exact amounts of memory in terms of the number of parameters and notice active GPU in practice to be roughly proportional to these counts. All experiments can run on a single 24GB GPU. We will report the exact numbers in the final manuscript.
>
> The Diff. Laplace objective performs better than MAP but is ~1.2-2x slower in training. We would like to mention that we rely on modern linearised Laplace approximations, which are currently actively being studied and improved. For instance, concurrent work [4] shows a 10x improvement of the Laplace approximation and should also readily be applicable to our setting.
>
> > MAP performing worse than the proposed Diff. Laplace method?
>
> Although MAP can very successfully fit model parameters (θ), it can typically not learn hyper-parameters (η), such as prior variances or symmetry constraints (in our case: layer-wise equivariances). The reason that such a data fitting objective does not encourage symmetry is that symmetries constrain the functions that a network can represent. Consequently, the regular objective that maximises train data fit will always prefer as little as possible equivariance (no symmetry, so σ=0), even when more equivariance would be preferable in terms of better generalisation on validation/test data. This is why in Deep Learning it is common to select hyper-parameters using validation data, e.g. cross-validation, or more complex with RL outer loops [3] or differentiated validation losses [8].
>
> Recent work by [5] shows that the marginal likelihood can be used to learn invariances directly on training data and demonstrated in [6] for DL. Unlike regular MAP, this objective has a built-in 'Occam's razor' effect and therefore balances good 'data fit' with model complexity (through the log determinant term). Similarly, we use Diff. Laplace to perform approximate Bayesian model selection. Crucially, we introduce a scalable parameterisation that allows differentiable layer-wise equivariances, unlike prior works that only consider invariances. We demonstrate that - in line with theory - Diff. Laplace allows us to optimise the right layer-wise equivariances η using training data. This demonstrates the principle of automatically learning layer-wise equivariances from training data.
>
> We hope that this shows that problems with the MAP objective are well-established facts. A more thorough discussion can be found in [5] and [6] and explicit examples that MAP cannot learn symmetry constraints in Sec. 4.2 of [7] and App. C of [6].
>
> [1] Benton, Gregory, et al. "Learning invariances in neural networks from training data." NeurIPS 2020
>
> [2] Wang, Rui, et al. "Approximately equivariant networks for imperfectly symmetric dynamics." ICML 2022
>
> [3] Cubuk, Ekin D., et al. "Autoaugment: Learning augmentation strategies from data." CVPR 2019
>
> [4] Immer, Alexander, et al. "Stochastic marginal likelihood gradients using neural tangent kernels." ICML 2023
>
> [5] Wilk, Mark van der, et al. "Learning invariances using the marginal likelihood." NeurIPS 2018
>
> [6] Immer, Alexander, et al. "Invariance learning in deep neural networks with differentiable Laplace approximations." NeurIPS 2022
>
> [7] van der Ouderaa, Tycho FA, et al. "Learning invariant weights in neural networks." UAI 2022
>
> [8] Lorraine, Jonathan, et al. "Optimizing millions of hyperparameters by implicit differentiation." AISTATS 2020

---

> > ### Comment · Reviewer_iYmm · 2023-08-12
> > **Response to the Rebuttal**
> >
> > I appreciate the authors for their prompt response and for clarifying the confusion regarding the comparison with the regular equivariant model. I have now updated my evaluation.

---

### Decision · Program_Chairs · 2023-09-21

**Decision:**

Accept (spotlight)

**Comment:**

All reviewers support the acceptance of this paper and the AC agrees with this evaluation. Overall, it is a well motivated paper on a challenging problem. A side note on additional references [A,B] that are very related to this work.

[A] Yeh, Raymond A., et al. "Equivariance discovery by learned parameter-sharing." International Conference on Artificial Intelligence and Statistics. PMLR, 2022.

[B] Yang, Jianke et al. “Generative Adversarial Symmetry Discovery.” International Conference on Machine Learning (ICML), 2023